# WHERE IS MOTION FROM? SCALABLE MOTION ATTRIBUTION FOR VIDEO GENERATION MODELS

## ABSTRACT

Despite the rapid progress of video generative models, the role of data in shaping motion is poorly understood. We present `Motive` (MOtion Training Influence for Video gEneration), a motion-centric, gradient-based data attribution framework that scales to modern, large, high-quality video datasets and models. We use this to study which fine-tuning clips improve or degrade temporal dynamics. `Motive` isolates temporal dynamics from static appearance via motion-weighted loss masks, yielding efficient and scalable motion-specific influence computation. On text-to-video models, `Motive` identifies clips that strongly affect motion and guides data curation that improves temporal consistency and physical plausibility. With `Motive`-selected high-influence data, our method improves both motion smoothness and dynamic degree on VBench, achieving a 74.1% human preference win rate compared with the pretrained base model. To our knowledge, this is the first framework to attribute motion rather than visual appearance in video generative models and to use it to curate fine-tuning data.

## 1 INTRODUCTION

Motion is the defining element of videos. Unlike image generation, which produces a single frame, video generative models capture how objects move, interact, and obey physical constraints (Wiedemer et al., 2025). Yet even with the rapid progress of video generative models, a fundamental question remains:

> *Which training clips influence the motion in generated videos?*

**Why it matters.** Diffusion models are data-driven, and their progress has tracked the scaling of data and compute (Saharia et al., 2022; Nichol & Dhariwal, 2021; Ho et al., 2022; Peebles & Xie, 2023). Prior work (Blattmann et al., 2023; Kaplan et al., 2020; Ravishankar et al., 2025) shows that training data shapes key generative properties, including visual quality (Rombach et al., 2022), semantic fidelity (Namekata et al., 2024), and compositionality (Wu et al., 2024b; Favero et al., 2025). Motion is no exception. *Motion* refers to temporal dynamics captured by optical flow, including trajectories, deformations, camera movement, and interactions. If generated motion reflects the data distribution that shaped the model, then attributing motion to influential training clips provides a direct lens on why a model moves the way it does and enables targeted data selection for desired dynamics.

High-quality data often matters most in fine-tuning, where large pretraining corpora are inaccessible and carefully chosen clips can have an outsized impact. Motion-specific attribution is therefore especially valuable in the fine-tuning regime, where the goal is to identify which clips most influence temporal coherence and physical plausibility.

**Why existing data attribution approaches are limited for motion.** Prior diffusion data attribution focuses on images and explains static content. Extending these methods to videos naïvely collapses motion into appearance, missing the temporal structure that distinguishes videos from images. Three challenges drive this gap: (i) localizing motion so attribution focuses on dynamic regions rather than static backgrounds, (ii) scaling to sequences since gradients must integrate across time, and (iii) capturing temporal relations like velocity, acceleration, and trajectory coherence that single-frame attribution cannot measure. Addressing motion attribution requires methods that explicitly model temporal structure, rather than treating time as an additional spatial axis.

**Our method.** We introduce `Motive`, a motion attribution framework for video diffusion models that isolates motion-specific influence. `Motive` computes gradients with motion-aware masking. As

a result, the attribution signal emphasizes dynamic regions rather than static appearance. Efficient approximations make the method practical for large, high-quality datasets and video generative models. The resulting scores trace generated motion back to training clips, enabling targeted curation and improving motion quality when used to guide fine-tuning.

**Our contributions are:**

1. Proposing a scalable gradient-based attribution approach for video generative models that is computationally efficient, even at the scale of modern, high-quality datasets and large generative models (§3.2).
2. Addressing a video-specific bias by correcting frame-length effects in gradient magnitudes, ensuring fair attribution across clips of different durations (§3.3).
3. Introducing an attribution that emphasizes temporal dynamics to trace which training clips most strongly influence motion quality (§3.4).
4. Showing that we improve motion smoothness and dynamic degree on VBench and in human trials (§4), matching - or surpassing - full-dataset fine-tuning performance with only 10% of the data, and outperforming motion-unaware attribution baselines (Tables 1 and 2).

## 2 BACKGROUND

A table of notation is in App. §A, as well as an extended related work in App. §B.

### 2.1 VIDEO GENERATION WITH DIFFUSION AND FLOW-MATCHING MODELS

**Diffusion and flow matching in latent space.** Let $p_\theta(\mathbf{v} \mid \mathbf{c})$ be a conditional generator with parameters $\theta$, where $\mathbf{v} \in \mathbb{R}^{F \times H \times W \times 3}$ is a clip of height $H$, width $W$, and $F$ frames, and $\mathbf{c}$ denotes conditioning such as text or other multimodal metadata (e.g., fps, depth, pose). We operate in VAE latents: $\mathbf{h} = E(\mathbf{v})$ and train a denoiser or velocity field on noisy latents. A noise scheduler supplies time-dependent coefficients $(\alpha_t, \sigma_t)$ controlling signal and noise scales, and the forward noising is:

$$\mathbf{z}(t, \boldsymbol{\epsilon}) \;=\; \alpha_t\, \mathbf{h} \;+\; \sigma_t\, \boldsymbol{\epsilon}, \quad \boldsymbol{\epsilon} \sim \mathcal{N}(0, \mathbf{I}), \quad t \in \{1, \dots, T\}. \tag{1}$$

*Denoising diffusion* (Ho et al., 2020) trains a network $\boldsymbol{\epsilon}_\theta(\mathbf{z}, \mathbf{c}, t)$ to predict the injected noise:

$$\mathcal{L}_{\text{diff}}(\theta; \mathbf{v}, \mathbf{c}) \;=\; \mathbb{E}_{t, \boldsymbol{\epsilon}}\Big[\|\boldsymbol{\epsilon}_\theta(\mathbf{z}(t, \boldsymbol{\epsilon}), \mathbf{c}, t) - \boldsymbol{\epsilon}\|_2^2\Big]. \tag{2}$$

*Flow matching* (Lipman et al., 2022; Albergo et al., 2023) learns a time-dependent vector field $\mathbf{f}_\theta(\mathbf{z}_t, \mathbf{c}, t)$ that matches the instantaneous velocity $\dot{\mathbf{z}} = \frac{d}{dt}\mathbf{z}$ induced by a chosen interpolant:

$$\mathcal{L}_{\text{flow}}(\theta; \mathbf{v}, \mathbf{c}) \;=\; \mathbb{E}_{t, \boldsymbol{\epsilon}}\Big[\|\mathbf{f}_\theta(\mathbf{z}(t, \boldsymbol{\epsilon}), \mathbf{c}, t) - \dot{\mathbf{z}}(t, \boldsymbol{\epsilon})\|_2^2\Big]. \tag{3}$$

Both objectives train time-indexed predictors over the latent space by integrating over $t$ and $\boldsymbol{\epsilon}$, thus gradient-based methods like attribution share similar challenges.

**From images to video for generation.** Adding a temporal axis materially changes modeling and training. Generation must capture spatial appearance and temporal dynamics such as object and camera motion, deformations, and interactions. Modern systems extend image backbones with temporal capacity, for example, 3D U-Nets or 2D U-Nets augmented with temporal attention, causal or sliding-window context, and factorized space-time blocks, often trained in a latent-video VAE that compresses frames while preserving temporal cues. Training departs from images along several axes, which we address in §3: (i) *Compute and storage.* Longer sequences multiply the cost of sampling timesteps, noise draws, and frames, motivating fixed-timestep or small-subset estimators that reduce variance without prohibitive cost (§3.2). (ii) *Variable horizon.* Clips vary in $F$ and frame rate (§3.3). (iii) *Time-specific failure modes.* Typical artifacts include inconsistent trajectories, temporal flicker, identity drift, and physically implausible dynamics despite sharp individual frames (§3.4).

**Motion representations in videos.** We denote our video as $\mathbf{v} = [\mathbf{f}_f]_{f=1}^F$ with $\mathbf{f}_f \in \mathbb{R}^{H \times W \times 3}$ being the $f$-th frame. We represent motion via optical flow between consecutive frames: $\mathbf{F}_f : \{1, \dots, H\} \times \{1, \dots, W\} \to \mathbb{R}^2$, where each flow vector in $\mathbb{R}^2$ encodes the horizontal displacement $\mathrm{d}w$ and vertical displacement $\mathrm{d}h$ of a pixel. The motion magnitude is $M_f(h, w) = \|\mathbf{F}_f(h, w)\|_2$. The $M_f$ over frames $f$ and pixels $h, w$ summarizes the amount and spatial layout of motion in a clip, which we will use to provide masks in our motion-weighted loss in §3.

## 2.2 DATA ATTRIBUTION

Data attribution measures how individual training samples affect a model's predictions. A classic approach to data attribution is to use influence functions (Koh & Liang, 2017). Intuitively, the influence of a training sample measures: if we upweight this training example, how much would the model's prediction on a test datum change? Consider a loss function $\mathcal{L}(\boldsymbol{\theta}; \mathbf{x})$ and a test sample $\mathbf{x}_{\text{test}}$, the influence of a training point $\mathbf{x}_n$ can be quantified as:

$$I(\mathbf{x}_n, \mathbf{x}_{\text{test}}) = -\nabla_{\boldsymbol{\theta}}\mathcal{L}(\boldsymbol{\theta}; \mathbf{x}_{\text{test}})^\top \mathbf{H}_{\boldsymbol{\theta}}^{-1} \nabla_{\boldsymbol{\theta}}\mathcal{L}(\boldsymbol{\theta}; \mathbf{x}_n), \quad \mathbf{H}_{\boldsymbol{\theta}} = \frac{1}{N}\sum_{n=1}^{N}\nabla_{\boldsymbol{\theta}}^2\mathcal{L}(\boldsymbol{\theta}; \mathbf{x}_n), \quad (4)$$

where the inverse Hessian captures the curvature of the loss landscape, yet computing or storing it is infeasible at modern model and dataset scales. Thus, practical methods (e.g., TracIn (Pruthi et al., 2020) and TRAK (Park et al., 2023)) approximate influence via gradient inner products or gradient feature projections.

**Attribution in diffusion models.** Diffusion training aggregates gradients over timesteps $t$ and noise draws $\boldsymbol{\epsilon}$, and gradient norms vary systematically with $t$. This produces a timestep bias where examples aligned with large-norm timesteps appear spuriously influential. Diffusion-ReTrac (Xie et al., 2024) reduces this bias by normalizing gradients and sub-sampling $t$ and $\boldsymbol{\epsilon}$ when forming influence. Let $\mathcal{L}_{\text{diff}}$ denote the diffusion loss, and with the sampled-timestep-and-noise set $\mathcal{T}$, we compute a cosine-style score between normalized test and train gradients:

$$I_{\text{diff}}(\mathbf{x}_n, \mathbf{x}_{\text{test}}) = \underbrace{\frac{1}{|\mathcal{T}_{\text{test}}|}\sum_{t,\boldsymbol{\epsilon}\in\mathcal{T}_{\text{test}}}\frac{\nabla_{\boldsymbol{\theta}}\mathcal{L}_{\text{diff}}(\boldsymbol{\theta}; \mathbf{x}_{\text{test}}, t, \boldsymbol{\epsilon})}{\|\nabla_{\boldsymbol{\theta}}\mathcal{L}_{\text{diff}}(\boldsymbol{\theta}; \mathbf{x}_{\text{test}}, t, \boldsymbol{\epsilon})\|}^\top}_{\text{normalized test gradients}} \underbrace{\frac{1}{|\mathcal{T}_n|}\sum_{t,\boldsymbol{\epsilon}\in\mathcal{T}_n}\frac{\nabla_{\boldsymbol{\theta}}\mathcal{L}_{\text{diff}}(\boldsymbol{\theta}; \mathbf{x}_n, t, \boldsymbol{\epsilon})}{\|\nabla_{\boldsymbol{\theta}}\mathcal{L}_{\text{diff}}(\boldsymbol{\theta}; \mathbf{x}_n, t, \boldsymbol{\epsilon})\|}}_{\text{normalized training gradients}}. \quad (5)$$

Averaging gradients over $(t, \boldsymbol{\epsilon})$ stabilizes estimates, and normalization mitigates timestep-induced scale effects. Attribution quality is also sensitive to the measurement function used to score examples, such as denoising loss versus likelihood proxies (Zheng et al., 2023).

**Why vanilla attribution is insufficient for videos.** Naïvely applying gradient-based attribution to video diffusion risks treating appearance and motion alike, often overemphasizing low-level appearance matches (objects, textures, backgrounds) while overlooking dynamics (Park et al., 2025; Tulyakov et al., 2018). Its cost grows with clip length, sampled timesteps, noise draws, and gradient dimensionality, making naïve methods impractical at modern video scales. Because we aim to explain and improve motion, we need attribution that suppresses static appearance, emphasizes motion-specific signals, and remains efficient, motivating the motion-centric approach in §3. Motion is distributed across frames and temporal horizons and entangled with static cues, so influence cannot be assigned by considering frames independently.

## 3 METHOD

We formalize the problem setup in §3.1 and develop a practical framework for motion attribution in video diffusion models with three key components: scalable gradient computation (§3.2), frame-length bias fix (§3.3), motion-aware weighting (§3.4) and data selection for targeted fine-tuning (§3.5). We also provide a computational efficiency analysis (§3.6) demonstrating the scalability of our approach to billion-parameter models and large-scale video datasets.

### 3.1 PROBLEM FORMULATION

We study data attribution for motion in the fine-tuning setting. Let $\mathcal{D}_{\text{ft}} = \{(\mathbf{v}_n, \mathbf{c}_n)\}_{n=1}^{N}$ be the fine-tuning corpus. Given a query video $(\hat{\mathbf{v}}, \hat{\mathbf{c}})$, we assign to each training clip $(\mathbf{v}_n, \mathbf{c}_n)$ a motion-aware influence score $I(\mathbf{v}_n, \hat{\mathbf{v}}; \boldsymbol{\theta})$ that explains how it contributes to the dynamics observed in $\hat{\mathbf{v}}$. The score should satisfy: (i) predictivity: rankings correlate with observed changes from fine-tuning on the most influential subsets; (ii) efficiency: scales to modern video generators, such as forgoing explicit Hessian inversion, expensive per-data integration, or prohibitive storage. To do this, we augment the influence target defined in Eq. 5 to be (a) lower variance for stable rankings with feasible levels of compute, (b) more scalable to store, and (c) motion-centric.

**fine-tuning Subset Selection.** For a budget $K \ll N$, we get a motion-influential subset by ranking scores and taking the top-$K$ examples. When aggregating across multiple query motions, we combine selections as described in §3. The resulting subsets serve as candidates for motion-centric fine-tuning.

## 3.2 SCALABLE GRADIENT-BASED ATTRIBUTION FOR GENERATIVE MODELS

We make attribution practical for modern, large, high-quality video datasets and models via inverse-Hessian approximations, lower-variance gradient-similarity estimators, low-cost single-sample estimators, and a Fastfood projection for tractable storage.

**Approximating the inverse-Hessian.** Computing exact inverse-Hessian-vector products is infeasible for modern neural networks. We estimate influence via gradient similarity, using an identity preconditioner for the inverse Hessian (Koh & Liang, 2017; Pruthi et al., 2020; Park et al., 2023).

**Common randomness for stable rankings.** To reduce variance without changing the target, we evaluate train and test gradients under the same $(t, \epsilon)$ pairs and average over a small set $\mathcal{T}$ (Xie et al., 2024; Lin et al., 2024). This paired averaging stabilizes rankings compared to independent draws:

$$I_{\text{diff}}^1(\mathbf{x}_n, \mathbf{x}_{\text{test}}) = \frac{1}{|\mathcal{T}|} \sum_{t, \epsilon \in \mathcal{T}} \underbrace{\frac{\nabla_{\boldsymbol{\theta}} \mathcal{L}_{\text{diff}}(\boldsymbol{\theta}; \mathbf{x}_{\text{test}}, t, \epsilon)}{\left\| \nabla_{\boldsymbol{\theta}} \mathcal{L}_{\text{diff}}(\boldsymbol{\theta}; \mathbf{x}_{\text{test}}, t, \epsilon) \right\|}}_{\text{normalized test gradients}}^{\top} \underbrace{\frac{\nabla_{\boldsymbol{\theta}} \mathcal{L}_{\text{diff}}(\boldsymbol{\theta}; \mathbf{x}_n, t, \epsilon)}{\left\| \nabla_{\boldsymbol{\theta}} \mathcal{L}_{\text{diff}}(\boldsymbol{\theta}; \mathbf{x}_n, t, \epsilon) \right\|}}_{\text{normalized training gradients}}. \tag{6}$$

**Single-sample variant for reduced compute.** We then fix a single $t_{\text{fix}}$ and a single shared draw $\epsilon_{\text{fix}} \sim \mathcal{N}(0, \mathbf{I})$ for all train–test pairs at the final checkpoint. Sharing $(t_{\text{fix}}, \epsilon_{\text{fix}})$ is key to having low enough variance, for the low-cost single-sample estimator to maintain relative ordering (Xie et al., 2024; Lin et al., 2024). The estimator collapses to:

$$I_{\text{diff}}^2(\mathbf{x}_n, \mathbf{x}_{\text{test}}) = \underbrace{\frac{\nabla_{\boldsymbol{\theta}} \mathcal{L}_{\text{diff}}(\boldsymbol{\theta}; \mathbf{x}_{\text{test}}, t_{\text{fix}}, \epsilon_{\text{fix}})}{\left\| \nabla_{\boldsymbol{\theta}} \mathcal{L}_{\text{diff}}(\boldsymbol{\theta}; \mathbf{x}_{\text{test}}, t_{\text{fix}}, \epsilon_{\text{fix}}) \right\|}}_{\text{normalized test gradient}}^{\top} \underbrace{\frac{\nabla_{\boldsymbol{\theta}} \mathcal{L}_{\text{diff}}(\boldsymbol{\theta}; \mathbf{x}_n, t_{\text{fix}}, \epsilon_{\text{fix}})}{\left\| \nabla_{\boldsymbol{\theta}} \mathcal{L}_{\text{diff}}(\boldsymbol{\theta}; \mathbf{x}_n, t_{\text{fix}}, \epsilon_{\text{fix}}) \right\|}}_{\text{normalized training gradient}}. \tag{7}$$

**Structured projection for reduced storage.** To operate at model scale, we apply a Johnson–Lindenstrauss projection via Fastfood (Le et al., 2014) and then normalize. Let

$$\mathbf{P} \in \mathbb{R}^{D' \times D} \quad \text{be implemented as} \quad \mathbf{P} := \frac{1}{\xi \sqrt{D'}} \mathbf{SQG\Pi QB}, \tag{8}$$

where $\mathbf{Q}$ is the Walsh–Hadamard matrix, $\mathbf{B}$ is a diagonal Rademacher matrix, $\mathbf{\Pi}$ is a random permutation, $\mathbf{G}$ is a diagonal Gaussian scaling, and $\mathbf{S}$ is a diagonal rescaling, and $\xi$ normalizes the variance. The projected, normalized gradient is:

$$\tilde{\mathbf{g}}(\boldsymbol{\theta}, \mathbf{x}) := \frac{\mathbf{P} \nabla_{\boldsymbol{\theta}} \mathcal{L}_{\text{diff}}(\boldsymbol{\theta}, \mathbf{x}, t_{\text{fix}}, \epsilon_{\text{fix}})}{\left\| \mathbf{P} \nabla_{\boldsymbol{\theta}} \mathcal{L}_{\text{diff}}(\boldsymbol{\theta}, \mathbf{x}, t_{\text{fix}}, \epsilon_{\text{fix}}) \right\|}. \tag{9}$$

Then the influence score is the compact cosine in $\mathbb{R}^{D'}$:

$$I_{\text{diff}}^3(\mathbf{x}_n, \mathbf{x}_{\text{test}}) = \underbrace{\tilde{\mathbf{g}}(\boldsymbol{\theta}; \mathbf{x}_{\text{test}})}_{\text{projected, normalized test gradient}}^{\top} \underbrace{\tilde{\mathbf{g}}(\boldsymbol{\theta}; \mathbf{x}_n)}_{\text{projected, normalized training gradient}}. \tag{10}$$

This keeps compute $\mathcal{O}(D' \log D')$ for projection and $\mathcal{O}(D')$ per dot product, with storage $\mathcal{O}(|\mathcal{D}| D')$, while staying close to the ranking behavior of full-gradient cosine similarity (Park et al., 2023).

## 3.3 VIDEO-SPECIFIC FRAME-LENGTH BIAS FIX

Raw gradient magnitudes depend on the number of frames $F$ in the video $\mathbf{v}$, thereby biasing scores toward longer videos. We correct this at measurement time by normalizing for frame count before the projection–normalization step:

$$\nabla_{\boldsymbol{\theta}} \mathcal{L}_{\text{diff}}(\boldsymbol{\theta}; \mathbf{v}, t_{\text{fix}}, \epsilon_{\text{fix}}) \leftarrow \frac{1}{F} \nabla_{\boldsymbol{\theta}} \mathcal{L}_{\text{diff}}(\boldsymbol{\theta}; \mathbf{v}, t_{\text{fix}}, \epsilon_{\text{fix}}). \tag{11}$$

We still apply $\ell_2$ normalization in Eq. 10, further stabilizing scales across examples. Together, single-timestep, common randomness, projection, and frame-length correction form a compact, scalable estimator that we use throughout. However, naïve video-level attribution conflates appearance with motion, often ranking clips high just because they share backgrounds or objects, while offering little insight into dynamics.

Figure 1: **Motive**. **Top.** Motion-gradient computation (§3.4) has three steps: (1) detect motion with AllTracker; (2) compute motion-magnitude patches; (3) apply loss-space motion masks to focus gradients on dynamic regions. **Bottom.** Our method (§3.2) is made scalable via a single-sample variant with common randomness and a projection, computed for each pair of training and query data, aggregated (§3.5) for a final ranking, and eventually used to select fine-tuning subsets.

## 3.4 MOTION ATTRIBUTION

To move beyond whole-video influence, we introduce motion attribution, which isolates the contribution of training data to temporal dynamics. Unlike video-level attribution, which treats each clip as a single unit and conflates appearance with motion, motion attribution reweights per-location gradients using motion masks, assigning influence via dynamic behavior rather than static content.

**Motion Masking Attribution.** Motion is what distinguishes video diffusion from image diffusion. Our goal is to understand how training data shapes motion in video diffusion models. Prior work has emphasized architectural or algorithmic changes for motion modeling (Peebles & Xie, 2023; Blattmann et al., 2023; Guo et al., 2023), many of the largest generative gains have instead come from scaling and curating massive video corpora, which in turn enable impressive motion synthesis results in video diffusion models (Ho et al., 2022; Wan et al., 2025; Tan et al., 2024; Yang et al., 2024). yet we lack tools that quantify how specific training clips shape particular motion patterns. We address this by attributing motion back to data via motion-weighted gradients, which yields actionable signals for targeted data selection, artifact diagnosis, and selective fine-tuning.

**Motion Detection and Latent Space Mapping.** Given a video $\mathbf{v} \in \mathbb{R}^{F \times H \times W \times 3}$ with $F$ frames of resolution $H \times W$, we first encode it into the VAE latent space as $\mathbf{h} = E(\mathbf{v}) \in \mathbb{R}^{F \times H/s \times W/s \times C}$, with downsampling factor $s = 8$ and $C = 16$ following the `Wan2.1` backbone used in our experiments. For motion computation, we use AllTracker (Harley et al., 2025) to extract motion information in pixel space: $A = \mathcal{A}(\mathbf{v}) \in \mathbb{R}^{F \times H \times W \times 4}$, where the first two channels contain optical flow maps $A_{:,:,:,0:2}$ indicating pixel displacement between frames, and the remaining channels $A_{:,:,:,2:4}$ encode visibility and confidence scores. We extract displacement vectors at each pixel location as:

$$\mathbf{D}_f(h, w) = (A_{f,h,w,0}, A_{f,h,w,1}) = (\mathrm{d}w, \mathrm{d}h). \tag{12}$$

We then bilinearly downsample motion quantities from $(H, W)$ to the latent grid $\left(\frac{H}{s}, \frac{W}{s}\right)$ so that our masking lives where gradients are computed.

**Motion-Weighted Gradient Computation.** We define the motion magnitude at each location as: $M_f(h, w) = \|\mathbf{D}_f(h, w)\|_2$. To obtain comparable motion weights across frames and pixels, we min–max normalize over all frames and pixels, ensuring values lie in $[0, 1]$:

$$\mathbf{W}(f, h, w) = \frac{M_f(h, w) - \min_{f', h', w'} M_{f'}(h', w')}{\max_{f', h', w'} M_{f'}(h', w') - \min_{f', h', w'} M_{f'}(h', w') + \zeta}, \tag{13}$$

where $\zeta = 10^{-6}$ ensures a positive denominator. This normalization mitigates bias from absolute motion scale, yielding weights that emphasize relative motion saliency rather than raw magnitude, following prior practice in video saliency detection (Fang et al., 2013). Let $(\tilde{h}, \tilde{w})$ index the latent grid. We obtain latent-aligned weights by bilinear downsampling:

$$\tilde{\mathbf{W}}(f, \tilde{h}, \tilde{w}) = \text{Bilinear}\big(\mathbf{W}(\cdot,\cdot,\cdot), F, \tfrac{H}{s}, \tfrac{W}{s}\big). \tag{14}$$

We compute per-location squared error at fixed $(t_{\text{fix}}, \boldsymbol{\epsilon}_{\text{fix}})$ at each frame $f$ and "latent pixel" $(\tilde{h}, \tilde{w})$:

$$\tilde{\mathcal{L}}_{\boldsymbol{\theta}, \mathbf{v}, \mathbf{c}}(f, \tilde{h}, \tilde{w}) = \Big( [\boldsymbol{\epsilon}_{\boldsymbol{\theta}}(\mathbf{z}(\mathbf{v}, t_{\text{fix}}, \boldsymbol{\epsilon}_{\text{fix}}), t_{\text{fix}}, \mathbf{c})]_{f, \tilde{h}, \tilde{w}} - [\boldsymbol{\epsilon}_{\text{target}}(t_{\text{fix}}, \boldsymbol{\epsilon}_{\text{fix}})]_{f, \tilde{h}, \tilde{w}} \Big)^2, \tag{15}$$

and define the motion-weighted loss by averaging over frames and latent spatial locations:

$$\mathcal{L}_{\text{mot}}(\boldsymbol{\theta}; \mathbf{v}, \mathbf{c}) = \frac{1}{F_{\mathbf{v}}} \text{mean}_{f, \tilde{h}, \tilde{w}} \Big[ \tilde{\mathbf{W}}_{\mathbf{v}, \mathbf{c}}(f, \tilde{h}, \tilde{w}) \cdot \tilde{\mathcal{L}}_{\boldsymbol{\theta}, \mathbf{v}, \mathbf{c}}(f, \tilde{h}, \tilde{w}) \Big]. \tag{16}$$

Notably, when $\tilde{\mathbf{W}}$ is all ones, this recovers the standard objective with no motion emphasis. The $1/F_{\mathbf{v}}$ factor corrects for frame-length bias and $F_{\mathbf{v}}$ signifies how the number of frames may be video-dependent. The corresponding motion-weighted gradient for attribution is:

$$I_{\text{mot}}(\mathbf{v}_n, \hat{\mathbf{v}}) = \tilde{\mathbf{g}}_{\text{mot}}(\boldsymbol{\theta}, \hat{\mathbf{v}})^{\top} \tilde{\mathbf{g}}_{\text{mot}}(\boldsymbol{\theta}, \mathbf{v}_n), \text{ where } \tilde{\mathbf{g}}_{\text{mot}}(\boldsymbol{\theta}, \mathbf{v}) := \frac{\mathbf{P}\mathbf{g}_{\text{mot}}(\boldsymbol{\theta}, \mathbf{v}, t_{\text{fix}}, \boldsymbol{\epsilon}_{\text{fix}})}{\|\mathbf{P}\mathbf{g}_{\text{mot}}(\boldsymbol{\theta}, \mathbf{v}, t_{\text{fix}}, \boldsymbol{\epsilon}_{\text{fix}})\|} \text{ and } \mathbf{g}_{\text{mot}} := \nabla_{\boldsymbol{\theta}} \mathcal{L}_{\text{mot}}. \tag{17}$$

Loss-space masking leaves forward noising and generation unchanged and reweights only attribution, avoiding interactions between motion weighting and noise injection. In contrast, our motion-aware attribution emphasizes dynamic regions and de-emphasizes static backgrounds, so rankings identify training clips that most strongly shape the model's motion rather than appearance.

### 3.5 MOST INFLUENTIAL FINE-TUNING SUBSET SELECTION

**Goal.** Given a query clip $(\hat{\mathbf{v}}, \hat{\mathbf{c}})$, we compute a motion-aware attribution value for each candidate fine-tuning example $(\mathbf{v}_n, \mathbf{c}_n) \in \mathcal{D}_{\text{ft}}$ using: $I_{\text{mot}}(\mathbf{v}_n, \hat{\mathbf{v}})$ from Eq. 17. Then, we construct a fine-tuning dataset $\mathcal{S}$ for one or many query videos $\hat{\mathbf{v}}$.

**Single-query-point fine-tuning selection.** For a budget of $K$ data points, we select the $K$ highest-scoring examples. In practice, $K$ is chosen as a percentile of the dataset size (e.g., top 1–10%), ensuring the subset scales consistently across datasets.

**Multi-query-point fine-tuning selection: aggregating attribution scores.** For $Q$ queries, we adopt the majority voting approach from ICONS (Wu et al., 2024a) and aggregate motion-aware influence scores across queries by percentile thresholding and voting. A sample receives a vote if the score is above the percentile cutoff $\tau$ for that query. The consensus score of a candidate $\mathbf{v}_n$ is the total number of queries that vote for it. We then rank all training samples by $\text{MajVote}(\mathbf{v}_n)$ and select the top-$K$ to form the fine-tuning subset. This formulation emphasizes samples that are consistently influential across multiple queries, without requiring cross-query calibration of raw scores.

$$\text{MajVote}_n = \sum_{q=1}^{Q} \mathbb{I}\big[ I_{\text{mot}}(\mathbf{v}_n, \hat{\mathbf{v}}_q) > \tau \big], \mathcal{S}_{\text{vote}}(K) = \big\{ \mathbf{v}_n | \mathbf{v}_n \text{ in top-}K \text{ by } \text{MajVote} \big\}. \tag{18}$$

### 3.6 COMPUTATIONAL EFFICIENCY ANALYSIS

**Gradient Compute.** Naïvely averaging over timesteps and noise for every example costs $\mathcal{O}(|\mathcal{D}| |\mathcal{T}| B)$, where $B$ is a single forward+backward cost and $|\mathcal{T}|$ is the number of sampled $t, \boldsymbol{\epsilon}$ per data. Using a single sample reduces this to $\mathcal{O}(|\mathcal{D}| B)$ – essential for having a reasonable cost on modern video datasets and models – while re-using the same sample across data allows the single-sample to have low enough variance for stable rankings. Projection adds $\mathcal{O}(D' \log D')$ per example using Fastfood (Le et al., 2014), negligible relative to a backward pass.

**Gradient Storage.** Storing full gradients is $\mathcal{O}(|\mathcal{D}| D)$. We instead store only projected vectors, $\mathcal{O}(|\mathcal{D}| D')$, plus the structured Fastfood state, $\mathcal{O}(D)$. Since $D'$ is typically orders of magnitude smaller than $D$, this transformation makes storage tractable for billion-parameter models.

**Data Ranking Compute.** Influence computation in Eq. 10 is an inner product in $\mathbb{R}^{D'}$, so evaluating all train examples against a query is $\mathcal{O}(|\mathcal{D}| D')$, and sorting is $\mathcal{O}(|\mathcal{D}| \log |\mathcal{D}|)$.

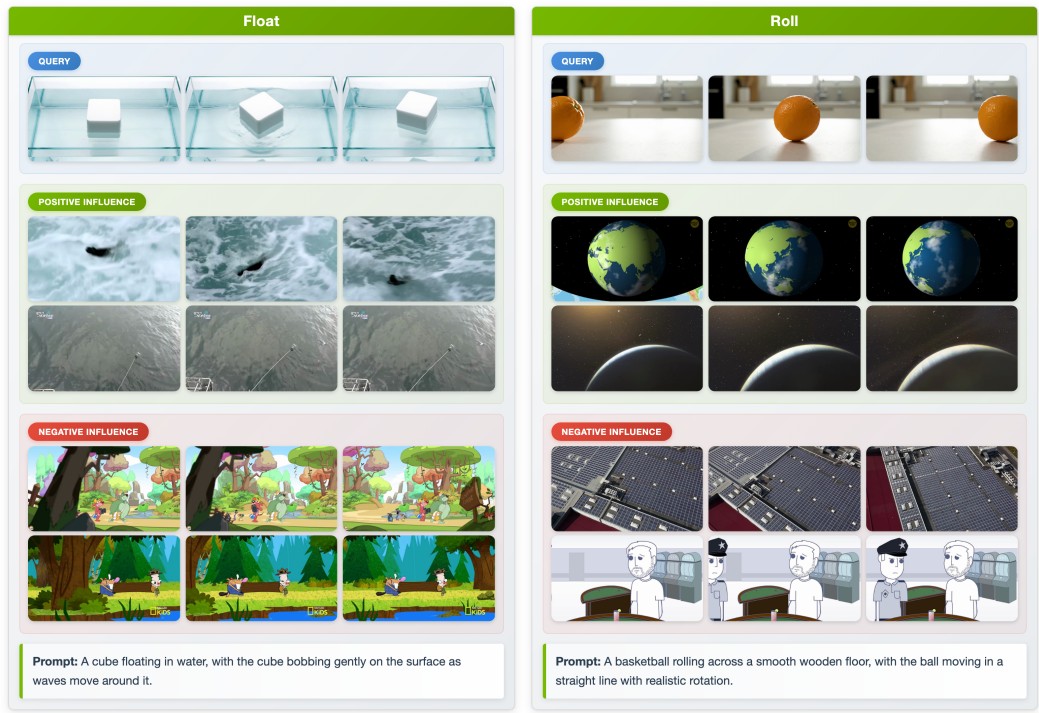

Figure 2: **Motion attribution examples.** *Top*: Query clips showing float (*left*) and roll (*right*) motions. *Middle*: Top-ranked positive training samples identified by `Motive` with high influence scores. *Bottom*: Negative influence samples with minimal motion, camera-only motion, or cartoon-style content that conflict with target motions.

**Additional Motion-Emphasis Compute.** Motion-specific overhead primarily stems from AllTracker mask extraction with complexity $\mathcal{O}(|\mathcal{D}| \cdot H \cdot W \cdot F)$ for clip length $F$ and frame resolution $H \times W$. Masks are extracted once, cached, and negligible relative to gradient cost. We provide a detailed runtime breakdown in App. H.1.

## 4 EXPERIMENT

### 4.1 SETUP

**fine-tuning Datasets.** We evaluate our motion attribution framework on two large-scale video datasets: `VIDGEN-1M` (Tan et al., 2024) and `4DNeX-10M` (Chen et al., 2025), both of which offer diverse motion patterns, rich temporal dynamics, and complex scenes. For our experiments, we use 10k videos from both datasets, which provide sufficient scale and diversity to thoroughly evaluate motion attribution methods across different temporal patterns and video-generation scenarios.

**Motion Query Data.** To evaluate our motion attribution, we curate a set of query videos representing distinct motion patterns and scenarios. Our query dataset consists of videos spanning multiple motion categories, with a focus on object dynamics: compress, bounce, roll, explode, float, free fall, slide, spin, stretch, swing. 5 videos, totaling 50 queries, represent each motion type. These videos are chosen for their clear, isolated motions, serving as a basis for evaluating attribution quality and downstream motion generation. Further details on query video curation are provided in App. F.2.

**Model & Baselines.** All experiments primarily use pretrained `Wan2.1-T2V-1.3B`, a widely used open-source baseline with strong performance and feasible compute. We further provide additional results on `Wan2.2-TI2V-5B` in App. C. Our baselines: **Base model** (pretrained, no fine-tuning); **Full fine-tuning** (approximate upper bound using the complete dataset); **Random selection** (uniform sampling); **Motion magnitude** (selects videos with the highest average motion magnitude); **Optical flow statistics** (selects videos with the highest composite scores based on motion magnitude, motion entropy, spatial coverage, motion complexity, and temporal consistency); **V-JEPA embeddings** (selects videos that are most representative of motion patterns using self-supervised spatiotemporal V-JEPA (Assran et al., 2025) features, which capture high-level motion semantics); and **Ours w/o motion masking** (influence of the entire video level without motion-specific masking);

| Method | Subject Consist. | Backgr. Consist. | Motion Smooth. | Dynamic Degree | Aesthetic Quality | Imaging Quality |
|---|---|---|---|---|---|---|
| Base | 95.3 | 96.4 | 96.3 | 82.3 | 45.3 | **65.7** |
| Full fine-tuning | 95.9 | **96.6** | 96.3 | 84.7 | 45.0 | 63.9 |
| Random selection | 95.3 | 96.6 | 96.3 | 81.6 | 45.7 | 65.1 |
| Motion magnitude | 95.6 | 96.2 | 95.7 | 82.0 | 45.1 | 63.2 |
| Optical flow | 95.3 | 96.4 | 95.3 | 80.0 | 45.4 | 62.9 |
| V-JEPA embedding | 95.7 | 96.0 | 95.6 | 82.0 | 44.9 | 62.7 |
| Ours w/o MM | 95.4 | 96.1 | 96.3 | 85.3 | 45.7 | 63.2 |
| **Ours (`Motive`)** | **96.3** | 96.1 | 96.3 | **89.4** | **46.0** | 64.6 |

Table 1: **VBench Evaluation.** Performance comparison on VBench (Huang et al., 2024) across different baselines (all values in %, higher is better). All selection methods use 10% of training data; our method uses majority vote aggregation (§3.5) across motion queries. MM: motion masking.

**Benchmark.** We evaluate our motion attribution framework with VBench (Huang et al., 2024), a video generation benchmark. VBench provides evaluation across dimensions, including subject and background consistency, motion smoothness, dynamic degree, and aesthetic and imaging quality. Since our method targets temporal dynamics, motion smoothness, and dynamic degree are most relevant; other metrics confirm that improvements do not sacrifice visual or semantic consistency.

**Implementation Details.** We finetune `Wan2.1-T2V-1.3B` with our `Motive`-selected high-quality video data following the official & DiffSynth-Studio[1] implementation. During fine-tuning, we update only the DiT backbone while freezing the T5 text encoder and VAE. All models are trained at a resolution of $480 \times 832$ with a learning rate of $1e{-}5$. Specialist models are trained on single motion category selected data, while generalist models use aggregated selections (both with top 10% selection from `VIDGEN-1M` (Tan et al., 2024) or `4DNeX` (Chen et al., 2025) with motion-weighted loss attribution). All training runs are conducted on 4-8 NVIDIA A100 GPUs. With one A100 GPU, it takes approximately 150 hours to compute the influence score of 10k samples. This process is highly parallelizable, on 64 GPUs, taking ~2.3 hours. The computed gradients can be reused for multiple selection queries, amortizing this one-time cost.

## 4.2 MAIN RESULTS

**High-influence selection and negative filtering.** Fig. 2 shows that motion-aware attribution ranks clips with clear, physically grounded dynamics and downranks those with little transferable motion. For rolling and floating, positives show continuous trajectories and smooth temporal evolution (turbulent water carrying objects; planetary rotation). Negatives are mostly static footage, camera-only motion, or cartoon clips whose simplified kinematics do not transfer to natural scenes. Our procedure promotes informative motions and filters data that would dilute temporal learning during fine-tuning. These trends hold across categories and align with the quantitative gains that follow.

**Qualitative improvements across motion types.** Fig. 3 compares the base pretrained model, naïve motion fine-tuning, and our motion-aware data selection for fine-tuning across four scenarios. Top: rubber-ball compression and coin spinning. Bottom: coffee mug sliding and red ball drop. Our method yields higher motion fidelity and temporal consistency than both baselines, especially for complex deformation, rotational dynamics, and physics-driven motion.

**Quantitative Results.** We evaluate our approach across different metrics using VBench (Huang et al., 2024), demonstrating consistent improvements in motion fidelity when fine-tuning with attribution-selected data compared to random sampling or naïve approaches. As shown in Tab. 1, `Motive` achieves the highest dynamic degree score (89.4%), significantly outperforming random selection (81.6%) and whole video attribution (85.3%). Our method also excels in subject consistency (96.3%) and aesthetic quality (46.0%), while maintaining competitive motion smoothness (96.3%). Notably, using only 10% of the training data, our approach surpasses the full finetuned model on dynamic degree (84.7%) and subject consistency (95.9%), demonstrating the superior empirical performance of motion-specific attribution for targeted fine-tuning. We further analyze the motion magnitude distribution of selected videos in App. D.

---

[1]https://github.com/modelscope/DiffSynth-Studio

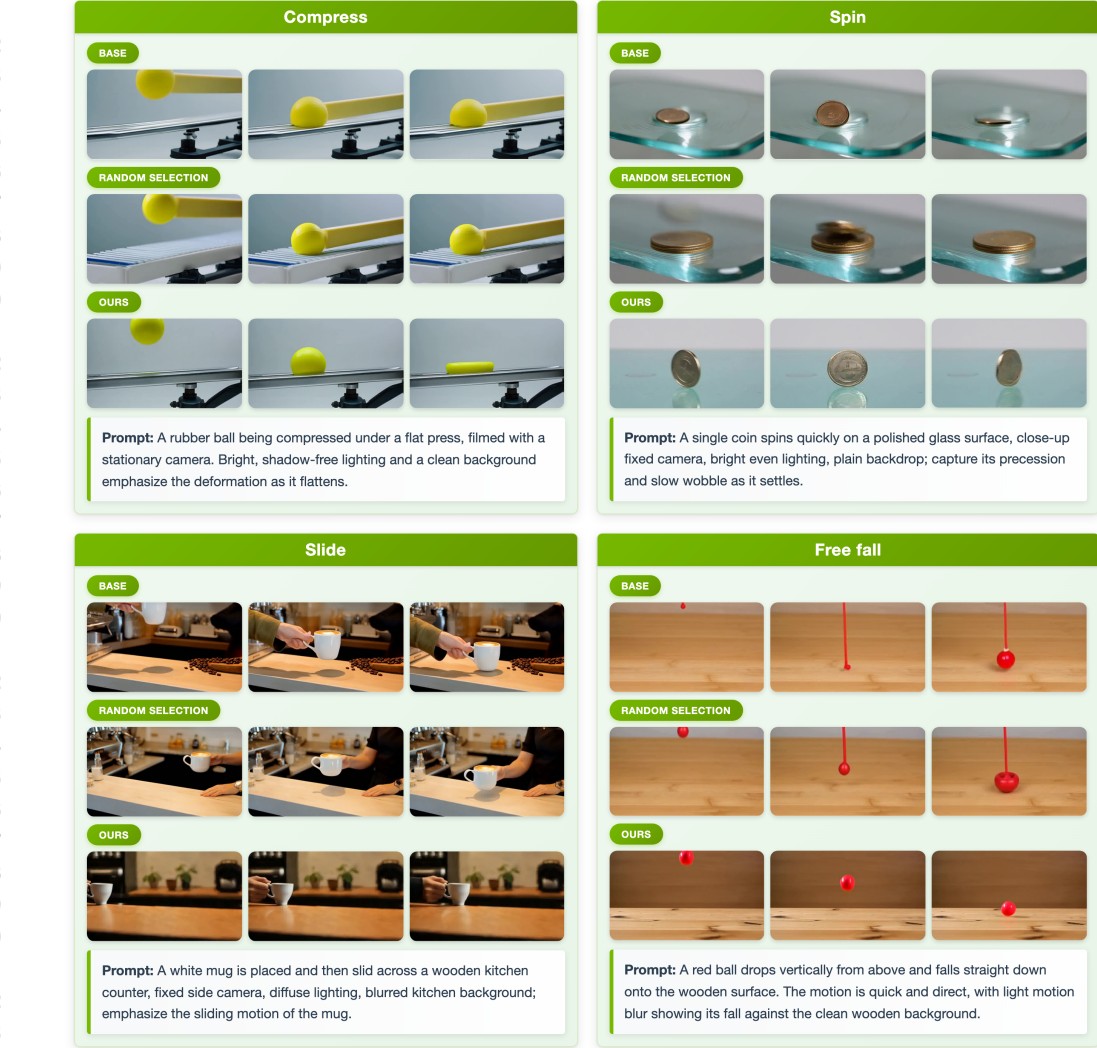

Figure 3: **Qualitative Comparisons.** We compare four motion scenarios (compress, spin, slide, free fall) across the base model, random selection, and our method. Our approach yields more realistic motion dynamics. Supplementary videos are included.

## 4.3 HUMAN EVALUATION

Automated scores can miss perceptual motion quality, so we run a human evaluation pairwise comparison protocol: participants view two generated videos and choose which shows better motion. We recruit 17 annotators and evaluate 10 motion categories. For each category, we prepare 5 test cases and compare our method to baselines across three pairings, yielding a balanced set of judgments. Presentation order is randomized, and ties are allowed. We report win rate (fraction our method is preferred), tie rate,

| Method | Win (%) | Tie (%) | Loss (%) |
|---|---|---|---|
| Ours vs. Base | 74.1 | 12.3 | 13.6 |
| Ours vs. Random | 58.9 | 12.1 | 29.0 |
| Ours vs. Full FT | 53.1 | 14.8 | 32.1 |
| Ours vs. w/o MM | 46.9 | 20.0 | 33.1 |

Table 2: **Human evaluation.** Pairwise comparisons across 50 videos with 17 participants (850 total). Win, tie, and loss rates show where our method is preferred, rated equal, or outperformed.

and overall preference. As shown in the table, annotators favor our attribution-guided selection: $74.1\%$ win rate vs. the base model and $53.1\%$ vs. the full finetuned model, indicating perceptually meaningful motion improvements.

## 4.4 ABLATIONS

**Single-timestep attribution.** Using a single timestep avoids the cost of averaging across timesteps while closely matching the multi-timestep baseline. With a fixed $t = 500$, we obtain $\rho = 68\%$

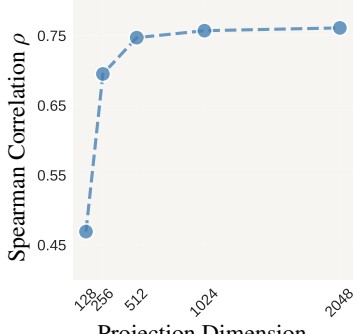

Figure 5: **Impact of Frame-Length Normalization on Motion Attribution.** Comparison of top-ranked samples for floating motion query. **Left**: With proper frame-length normalization, top samples consistently exhibit floating motion (waves, floating objects, surfing). **Right**: Without normalization, rankings are biased by video length, resulting in no coherent patterns among top samples.

agreement with the 7-timestep setting $t \in \{150, 300, 400, 500, 600, 750, 850\}$. Using the same timestep for both training and testing is key to preserving relative rankings. Early timesteps (e.g., 250) add little noise; late timesteps (e.g., 750) heavily corrupt inputs and can obscure motion cues. $t = 500$ strikes a balance, delivering high correlation and substantial compute savings. Averaging multiple timesteps yields minimal ranking gains, and incorporating late-timestep gradients can bias rankings. A single fixed timestep is therefore sufficient for variance-reduced, scalable attribution.

**Projected Gradients Preserve Influence Rankings.**

Comparing full gradients for attribution is infeasible at a billion-parameter scale. We reduce dimensionality with structured random projections that preserve influence geometry, ablating $D' \in \{128, \ldots, 2048\}$ against the full-gradient baseline. We assess ranking preservation via Spearman correlation with un-projected scores (Fig. 4). Small projections preserve rankings poorly: $D' = 128$ yields $\rho = 46.9\%$. Preservation improves with size: $D' = 512$ reaches $\rho = 74.7\%$. Beyond that, gains are marginal while cost rises: $D' = 1024$ ($\rho = 75.7\%$) and $D' = 2048$ ($\rho = 76.1\%$). Thus, $D' = 512$ offers the best trade-off, scaling to large models while maintaining quality.

**Frame-Length Normalization.** Following the Wan training protocol, we standardize all videos to 81 frames at 16 fps (satisfying the $4n+1$ constraint) to enable fair attribution across clips of different raw lengths. Without standardization, gradient-based scores correlate strongly with video length rather than motion quality ($\rho = 78.0\%$), leading to longer clips ranking higher regardless of dynamics. Standardizing frames reduces this spurious length correlation by $54.0\%$ while preserving motion-based correlation, so rankings reflect motion rather than duration. As in Fig. 5, normalization clarifies motion-specific patterns. For floating queries with frame-length normalization (left), top-ranked samples consistently show wave dynamics, floating objects, and surfing, all of which match the target motion. Without normalization (right), top samples lack coherent similarity because rankings are driven by clip length, hindering identification of motion-relevant training examples.

Figure 4: **Projection dimension analysis.** Spearman correlation between projected and full gradients shows rapid improvement with projection dimension, with 512 providing a strong trade-off between accuracy and efficiency.

## 5 CONCLUSION

We address a central and underexplored question in video diffusion: where is motion from? We propose `Motive` that traces generated dynamics back to influential training clips by isolating motion-specific gradients. Unlike image-based attribution, our method directly targets temporal dynamics, revealing how coherence and physical plausibility emerge from data. Our results show that motion learning is traceable to specific examples, providing a quantitative tool for diagnosing artifacts and enabling targeted data selection and curation. This enables more controllable and interpretable video diffusion models, and as models scale, such data-level understanding will be essential for building robust and reliable generative systems.

**Limitations.** (i) Motion saliency depends on the chosen tracker; severe occlusions or transparency can degrade masks. (ii) Camera-only motion and very subtle micro-motion remain challenging to separate without extra signals (e.g., camera pose). (iii) Our evaluation centers on one open-source backbone due to compute; broader portability is future work. Further discussion is in App. H.

## REPRODUCIBILITY STATEMENT

We provide clear descriptions and supporting resources to enable reproducibility of our motion attribution framework. Our method is detailed in §3, including motion-aware gradient computation, structured random projections, and influence score calculation. Experimental configurations, datasets, and evaluation protocols are thoroughly documented in §4, with additional algorithm and implementation details provided in App. §E and App. §F. We use publicly available video datasets and evaluate using established benchmarks, including VBench (Huang et al., 2024). Our human evaluation protocol, which includes pairwise comparisons, is described in detail along with the evaluation criteria. To facilitate replication and extension of our work, we will seek to release our codebase upon acceptance. These resources will enable researchers to reproduce our experiments and build upon our motion-aware data attribution approach.

## ETHICS STATEMENT

Our work focuses on developing methods for motion attribution in video diffusion models using publicly available datasets (VIDGEN-1M and 4DNeX-10M). We do not collect or use personally identifiable information or sensitive data. We also conduct small-scale human evaluations to assess perceptual motion quality, using voluntary participation without sensitive or identifiable data. The proposed framework improves transparency and controllability in generative models, which we believe has positive broader impacts. By exposing which data drives motion behaviors, our approach supports auditing and curation to avoid undesirable dynamics while preserving utility.

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

# Appendices

# A  NOTATION

Table 3: Glossary and notation.

| Symbol | Description |
|---|---|
| *Acronyms and Basic Notation* | |
| VAE | Variational Autoencoder |
| DiT | Diffusion Transformer backbone |
| $\mathbf{I}$ | Identity matrix |
| *Video Generation* | |
| $p_{\boldsymbol{\theta}}(\mathbf{v} \mid \mathbf{c})$ | Conditional video generator with parameters $\boldsymbol{\theta}$ |
| $\mathbf{v} \in \mathbb{R}^{F \times H \times W \times 3}$ | Video clip with frames $F$, height $H$, width $W$ |
| $\mathbf{c}$ | Conditioning signal such as text or multimodal metadata |
| $\boldsymbol{\theta}$ | Trainable model parameters |
| $f \in \{1, \ldots, F\}$ | Frame index |
| $h \in \{1, \ldots, H\}, w \in \{1, \ldots, W\}$ | Spatial indices for height and width respectively |
| *Datasets* | |
| $\mathcal{D} = \{(\mathbf{v}_n, \mathbf{c}_n)\}_{n=1}^{N}$ | Training corpus with size $N$ and index $n$ |
| $\mathcal{D}_{\mathrm{ft}} \subseteq \mathcal{D}$ | Fine-tuning dataset |
| $\mathcal{S} \subseteq \mathcal{D}$ | Selected influential subset |
| $k \in \{1, \ldots, K\}$ | The selected subset size |
| $q \in \{1, \ldots, Q\}$ | Number of query clips |
| $\hat{\mathbf{v}}, \hat{\mathbf{c}}$ | Query video and its conditioning |
| *Latent Space and Diffusion Components* | |
| $E$ | VAE encoder |
| $\mathbf{h} = E(\mathbf{v}) \in \mathbb{R}^{F \times (H/s) \times (W/s) \times C}$ | Latent video with spatial factor $s$ and channels $C$ |
| $\mathbf{z}$ | Noisy latent variable used in diffusion or flow matching |
| $\boldsymbol{\epsilon} \sim \mathcal{N}(0, \mathbf{I})$ | Gaussian noise |
| $\boldsymbol{\epsilon}_{\boldsymbol{\theta}}(\mathbf{z}, \mathbf{c}, t)$ | Predicted noise network in diffusion training |
| $\mathbf{f}_{\boldsymbol{\theta}}(\mathbf{z}, \mathbf{c}, t)$ | Time-indexed vector field in flow matching |
| $\dot{\mathbf{z}}$ | Time derivative of the latent trajectory |
| $\alpha_t, \sigma_t$ | Scheduler signal and noise scales at timestep $t$ |
| $\boldsymbol{\epsilon}_{\mathrm{target}}$ | Target noise or velocity used for supervision |
| $t \in \{1, \ldots, T\}$ | Diffusion or flow-matching timestep, with total timesteps $T$ |
| $t_{\mathrm{fix}}, \boldsymbol{\epsilon}_{\mathrm{fix}}$ | Fixed timestep and shared noise draw used for low-variance gradients |
| *Attribution and Influence* | |
| $I(\mathbf{v}_n, \hat{\mathbf{v}}; \boldsymbol{\theta})$ | Influence score between a train clip and a query clip |
| $I_{\mathrm{mot}}(\mathbf{v}_n, \hat{\mathbf{v}}; \boldsymbol{\theta})$ | Motion-aware influence score |
| $\mathrm{TopK}(\cdot)$ | Top-$K$ operator for selecting highest scores |
| $\mathrm{MajVote}(\cdot)$ | Majority-vote aggregation across queries |
| $\tau$ | Percentile cutoff for voting |
| $\rho$ | Spearman correlation coefficient |

Table 4: Glossary and notation (continued).

| Symbol | Description |
|---|---|
| *Loss Functions* | |
| $\mathcal{L}$ | Generic loss |
| $\mathcal{L}_{\text{diff}}(\boldsymbol{\theta}; \mathbf{v}, \mathbf{c}), \mathcal{L}_{\text{flow}}(\boldsymbol{\theta}; \mathbf{v}, \mathbf{c})$ | Diffusion and flow-matching objective |
| $\mathcal{L}_{\text{mot}}(\boldsymbol{\theta}; \mathbf{v}, \mathbf{c})$ | Motion-weighted objective used for attribution |
| $\tilde{\mathcal{L}}$ | Per-location squared error in latent space |
| $\mathbf{g}, \tilde{\mathbf{g}}$ | Gradient and its projected version |
| $\mathbf{g}_{\text{mot}}, \tilde{\mathbf{g}}_{\text{mot}}$ | Motion-weighted gradient and its projection |
| $\mathbf{H}_{\boldsymbol{\theta}}$ | Hessian with respect to $\boldsymbol{\theta}$ |
| *Motion Representations* | |
| $\mathcal{A}(\mathbf{v}) = A$ | AllTracker motion extraction |
| $A \in \mathbb{R}^{F \times H \times W \times 4}$ | Motion tensor containing flow, visibility, and confidence |
| $\mathbf{D}_f(h, w)$ | Displacement vector at frame $f$ and location $(h, w)$ |
| $M_f(h, w)$ | Motion magnitude at a location, computed from the displacement |
| $\mathbf{W}(f, h, w) \in [0, 1]$ | Normalized motion weights used to mask per-location losses |
| *Projections and Computational Details* | |
| $D, D'$ | Full and projected gradient dimensions |
| $\mathbf{P} \in \mathbb{R}^{D' \times D}$ | Projection matrix used for Fastfood-style JL projection |
| $\xi$ | Variance normalization constant for projection |
| $\mathcal{T}$ | Set of sampled $(t, \epsilon)$ pairs for gradient estimation |
| $B$ | Unit compute cost used in complexity accounting |

## B EXTENDED RELATED WORK

### B.1 DATA ATTRIBUTION

Understanding how individual training examples shape model behavior has been a long-standing goal in machine learning. Modern data attribution methods fall into two main groups (Hammoudeh & Lowd, 2024): retraining-based methods (e.g., leave-one-out (Cook et al., 1982; Jia et al., 2021), downsampling (also known as subsampling or counterfactual influence) (Feldman & Zhang, 2020), Shapley-value (Wang et al., 2024a;b)) and gradient-based methods (influence-function family, including Influence Functions (Koh & Liang, 2017), TracIn (Pruthi et al., 2020), and TRAK (Park et al., 2023)). Influence functions provide a principled framework by approximating the effect of removing a training point, TracIn (Pruthi et al., 2020) and TRAK (Park et al., 2023) make attribution feasible at scale. While effective for classification, these methods assume a direct mapping between training gradients and predictions, which becomes more complex in generative models.

Data attribution refers to methods that trace how individual training examples (or subsets) influence a model's predictions or behavior. Formally, it assigns an attribution score to each training sample, estimating the extent to which that sample contributes (positively or negatively) to the model's output on a given test query or behavior. Before diffusion models, attribution methods were applied in supervised learning tasks such as classification and regression, where influence functions (Koh & Liang, 2017) and scalable approximations like TracIn (Pruthi et al., 2020), TRAK (Park et al., 2023), and TDA (Bae et al., 2024) quantified the impact of training examples on downstream predictions. Recent work adapted data attribution to diffusion models, where iterative denoising introduces timestep-dependent bias. Mlodozeniec et al. (2024) propose scalable approximations, while Xie et al. (2024) identify timestep-induced artifacts and normalization schemes. Concept-TRAK (Park et al., 2025) extends attribution to concepts by reweighting gradients with concept-specific rewards, enabling attribution to semantic factors. Wang et al. (2023a) instead design a customization-based benchmark for text-to-image models, where models are fine-tuned on exemplar images with novel tokens and attribution is evaluated by whether it can recover the responsible exemplars. However, these works are limited to image diffusion, which captures static appearance but not temporal dynamics.

### B.2 MOTION IN VIDEO GENERATION

Video diffusion extends image generation to time, requiring coherent motion across frames (Ho et al., 2022; Blattmann et al., 2023; Peebles & Xie, 2023; Wan et al., 2025; Agarwal et al., 2025). A large body of work builds temporal structure via attention layers (Wu et al., 2023), control signals (Chen et al., 2023; Zhang et al., 2023), feature correspondences (Geyer et al., 2023; Bao et al., 2023; Wang et al., 2024c), or consistency distillation (Wang et al., 2023b; Zhou et al., 2024). Recent work has highlighted the challenge of decoupling motion from appearance in video diffusion transformers, where spatial and temporal information become entangled in the model's representations (Shi et al., 2025). However, understanding *which* training clips influence specific motion patterns in generated videos remains an open challenge.

In parallel, motion has long been studied through optical flow and correspondence – from classical formulations (Horn & Schunck, 1981; Lucas & Kanade, 1981) to modern deep flows like RAFT, which improves accuracy and generalization (Teed & Deng, 2020). These priors are often repurposed in generation for guiding dynamics or checking temporal consistency, but they do not explain *which* training examples shaped a model's motion behavior. Our work addresses both gaps by introducing a motion-aware data attribution framework specifically designed for video diffusion. We use motion-weighted gradients that disentangle temporal dynamics from static appearance, enabling us to trace generated motion patterns back to the most influential training clips.

## C ADDITIONAL EXPERIMENTS

### C.1 RESULTS ON DIFFERENT VIDEO GENERATION MODELS

We further test our framework on additional video generation architectures beyond Wan2.1-T2V-1.3B. We have included the experiment results of `Motive` on Wan2.2-TI2V-5B, which introduces a much

| Model | Method (↓) / Metric (→) | Subject Consist. | Background Consist. | Motion Smooth. | Dynamic Degree | Aesthetic Quality | Imaging Quality |
|---|---|---|---|---|---|---|---|
| **Wan2.2-T2V-5B** | Base | 94.9 | 96.4 | 97.5 | 84.0 | 44.4 | 65.5 |
| | Full fine-tuning | **95.3** | 96.5 | 97.5 | 87.5 | 44.8 | 66.2 |
| | Random selection | 94.7 | 96.2 | 97.3 | 83.8 | 44.6 | 65.2 |
| | Ours w/o MM | 94.9 | 96.5 | 97.4 | 86.2 | 45.2 | 64.8 |
| | **Ours (Motive)** | 95.1 | **96.6** | **97.6** | **91.0** | **45.6** | 65.5 |

Table 5: **VBench Evaluation on Additional Model.** Following the same setting in §4, we extend the VBench (Huang et al., 2024) evaluation to Wan2.2-T2V-5B, a larger-scale text-to-video model. Random selection and our **Motive** both select 10% of the training data, with our method using majority vote aggregation (§3.5) across all motion queries. Results demonstrate that **Motive** generalizes effectively to different models. MM: motion masking.

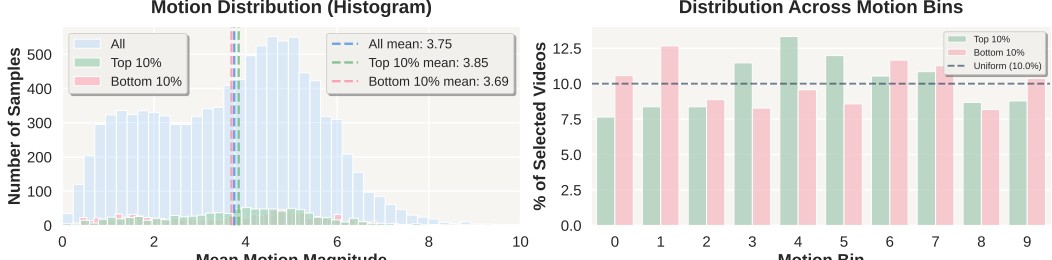

Figure 6: **Motive is not simply selecting "motion-rich" clips.** Our influence scores are computed via gradients, and training videos are considered influential only when they directly improve the model's ability to generate the target motion dynamics, not because they contain more motion overall. To empirically validate this, we analyze the distribution of motion magnitudes in our selected data. We compute the mean motion magnitude for 10k videos in the **VIDGEN** dataset and compare the distributions of the top 10% (highest influence scores) and the bottom 10% (lowest influence scores). Our top 10% selected videos have a mean motion magnitude of 3.85, which is only 4.3% higher than the bottom 10% (3.69), despite representing opposite extremes of influence scores. The analysis also shows that in the moderate motion range (bins 3, 4, and 5), the top 10% of positive-influence samples outnumber the bottom 10% of negative-influence samples. Yet, both groups also appear in low-motion bins (0-2) and high-motion bins (6-9). This distribution pattern shows that high-influence videos selected by **Motive** span the entire motion spectrum, not just high-motion regions. Many of the highest-motion videos receive low influence scores, while numerous influential videos exhibit modest or even low motion magnitude. These findings show that our motion attribution approach captures training influence, focusing on motion rather than simply acting as a motion-saliency filter.

larger parameter count (5B vs. 1.3B) and a new high-compression Wan2.2-VAE. The results in Tab. 5 show that our approach generalizes effectively across different model designs.

# D ANALYSIS

## D.1 MOTION DISTRIBUTION ANALYSIS

**Motive is not simply selecting "motion-rich" clips:** The key distinction is that our influence scores are computed via gradients, and training videos are considered influential only when they directly allow the model to lower the loss, improving the model's ability to generate the target motion dynamics, not because they contain more motion overall.

To empirically validate this, we further analyze the distribution of motion magnitudes in our selected data. We compute the mean motion magnitude for 10k videos in the **VIDGEN** dataset and compare the distributions of the top 10% (highest influence scores) and the bottom 10% (lowest influence scores).

Our top 10% selected videos have a mean motion magnitude of 3.85, which is only 4.3% higher than the bottom 10% (3.69), despite representing opposite extremes of influence scores. The analysis

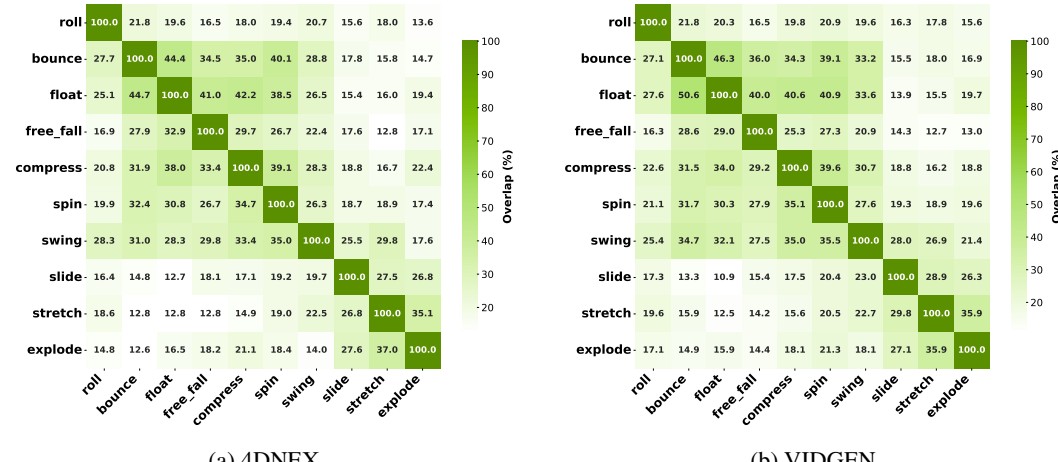

(a) 4DNEX  (b) VIDGEN

Figure 7: **Cross-motion influence overlap across datasets.** Heatmaps showing the percentage overlap of top-100 influential training samples across different motion categories for (a) 4DNEX and (b) VIDGEN datasets. Each cell $(i, j)$ represents the percentage of motion category $i$'s influential data (aggregated from 5 queries per category) that also appears in motion category $j$'s top-100 influential samples. The asymmetric nature of the matrices (e.g., bounce→float ≠ float→bounce) arises because different motion categories have different numbers of unique influential videos, leading to directional overlap percentages. Consistent high-overlap pairs (e.g., bounce-float: 44.4%/46.3%) and low-overlap pairs (e.g., free fall-stretch: 12.8%/12.7%) across datasets validate that these influence patterns reflect fundamental aspects of motion representation in video generation models.

also shows that in the moderate motion range (bins 3, 4, and 5), the top 10% of positive-influence samples outnumber the bottom 10% of negative-influence samples. Yet, both groups also appear in low-motion bins (0-2) and high-motion bins (6-9).

This distribution pattern shows that high-influence videos selected by `Motive` span the entire motion spectrum, not just high-motion regions. Many high-motion videos receive low influence scores, while numerous influential videos exhibit modest or even low motion magnitude. These findings show that our motion attribution approach captures training influence, focusing on motion rather than simply acting as a motion-saliency filter.

### D.2   Cross-Motion Influence Patterns

To analyze cross-motion influence patterns, we examine the percentage overlap of top-100 influential training data across different motion categories in both 4DNEX and VIDGEN datasets. As described in §4, we use 5 query samples to identify the top-100 most influential training videos, aggregating results across queries. As shown in Fig. 7, both datasets exhibit remarkably similar patterns with mean overlaps of 24.0% and 24.3%, respectively, indicating moderate sharing of influential data across motion categories.

Both datasets consistently identify the same high-overlap pairs: bounce-float (44.4%/46.3%), compress-float (40.1%/34.0%), and compress-spin (36.9%/39.6%), suggesting these motions share fundamental characteristics that the model learns from similar training examples. Conversely, low-overlap pairs such as free fall-stretch (12.8%/12.7%) and float-slide (14.0%/10.9%) indicate more specialized influential data for mechanically dissimilar motions. The influence matrices are asymmetric because the number of unique influential samples shared across the 5 query samples differs across motion categories. The similar cross-motion influence patterns observed across both the 4DNEX and VIDGEN datasets demonstrate that these relationships are generalizable across different video datasets and reflect dynamic similarity.

                    20                    

---

**Algorithm 1** `Motive`: Motion-Aware Data Attribution Framework

---

**Require:** fine-tuning corpus $\mathcal{D}_{\text{ft}} = \{(\mathbf{v}_n, \mathbf{c}_n)\}_{n=1}^N$, query video $(\hat{\mathbf{v}}, \hat{\mathbf{c}})$, fixed $(t_{\text{fix}}, \boldsymbol{\epsilon}_{\text{fix}})$, projection matrix $\mathbf{P}$

**Ensure:** Motion-aware influence scores $\{I_{\text{mot}}(\mathbf{v}_n, \hat{\mathbf{v}})\}_{n=1}^N$

1: **for** $(\mathbf{v}_n, \mathbf{c}_n) \in \mathcal{D}_{\text{ft}}$ **do**
2:     $A_n \leftarrow \text{ALLTRACKER}(\mathbf{v}_n)$                    ▷ Extract per-pixel flow displacements $\mathbf{D}_f$ (Eq. 12)
3:     Downsample and normalize to latent-space motion mask $\mathbf{W}_n$ (Eqs. 13–15)
4:     Evaluate motion-weighted loss $\mathcal{L}_{\text{mot}}(\boldsymbol{\theta}; \mathbf{v}_n, \mathbf{c}_n)$ (Eq. 16)
5:     Compute motion gradient $\mathbf{g}_{\text{mot}}(\boldsymbol{\theta}, \mathbf{v}_n, t_{\text{fix}}, \boldsymbol{\epsilon}_{\text{fix}}) = \nabla_{\boldsymbol{\theta}} \mathcal{L}_{\text{mot}}(\boldsymbol{\theta}; \mathbf{v}_n, \mathbf{c}_n, t_{\text{fix}}, \boldsymbol{\epsilon}_{\text{fix}})$
6:     Normalize by frame length: $\mathbf{g}_{\text{mot}} \leftarrow \mathbf{g}_{\text{mot}}/F$
7:     Project motion gradient: $\tilde{\mathbf{g}}_{\text{mot}}(\boldsymbol{\theta}, \mathbf{v}_n) := \frac{\mathbf{P}\mathbf{g}_{\text{mot}}(\boldsymbol{\theta}, \mathbf{v}_n, t_{\text{fix}}, \boldsymbol{\epsilon}_{\text{fix}})}{\|\mathbf{P}\mathbf{g}_{\text{mot}}(\boldsymbol{\theta}, \mathbf{v}_n, t_{\text{fix}}, \boldsymbol{\epsilon}_{\text{fix}})\|}$ (Eq. 17)
8: **end for**
9: Compute query gradient: $\tilde{\mathbf{g}}_{\text{mot}}(\boldsymbol{\theta}, \hat{\mathbf{v}})$ using the same procedure for $(\hat{\mathbf{v}}, \hat{\mathbf{c}})$
10: **for** $n = 1, \ldots, N$ **do**
11:     $I_{\text{mot}}(\mathbf{v}_n, \hat{\mathbf{v}}) = \tilde{\mathbf{g}}_{\text{mot}}(\boldsymbol{\theta}, \hat{\mathbf{v}})^\top \tilde{\mathbf{g}}_{\text{mot}}(\boldsymbol{\theta}, \mathbf{v}_n)$ (Eq. 17)
12: **end for**
13: Rank all training clips by $I_{\text{mot}}(\mathbf{v}_n, \hat{\mathbf{v}})$ and select top-$K$ influential samples using majority vote aggregation (Eq. 18):

$$\mathcal{S} = \mathcal{S}_{\text{vote}}(K) = \{\mathbf{v}_n | \mathbf{v}_n \text{ in top-}K \text{ by } \text{MajVote}\}$$

14: **return** $\mathcal{S}$

---

## E    ADDITIONAL METHOD DETAILS

**Tracker-agnostic scope.** We treat the motion estimator as a pluggable source of saliency rather than a training dependency. Given displacement magnitudes, we construct latent-space weights via bilinear mapping and normalization. Our implementation supports the use of alternative estimators (such as dense optical flow or point tracking) with identical interfaces, enabling practitioners to swap AllTracker without modifying the attribution code.

**Model-agnostic scope.** Our attribution only requires per-example gradients under matched $(t_{\text{fix}}, \boldsymbol{\epsilon}_{\text{fix}})$, and therefore applies to both diffusion and flow-matching objectives. The score reduces to a gradient inner product under a fixed preconditioner; the generator architecture affects gradient statistics but not the definition of influence. In practice, replacing the denoiser or velocity field leaves the weighting and aggregation unchanged.

**Algorithm Summary.** For completeness, Algorithm 1 summarizes the full `Motive` pipeline, detailing the computation of motion-weighted gradients, projection into low-dimensional space, and the subsequent influence-based ranking and selection of training clips.

**Rationale for synthetic queries.**    The query set is not used as training data; instead, it specifies targets for attribution and for multi-query aggregation. Synthetic generation offers controllability that is difficult to achieve at scale with web videos. This design yields near-realistic yet standardized stimuli aligned with our goal of probing motion-specific influence.

## F    ADDITIONAL EXPERIMENT DETAILS

### F.1    HYPERPARAMETER SETTINGS

For reproducibility, we document the hyperparameters used throughout attribution, subset selection, and fine-tuning. Where values were not explicitly tuned, we adopted defaults from DiffSynth-Studio and the official `wan` repo.

**Attribution.** Motion-aware influence estimation is computed at a single fixed timestep $t_{\text{fix}} = 500$, selected as a mid-range value that correlates strongly with multi-timestep averaging. A shared Gaussian draw $\boldsymbol{\epsilon}_{\text{fix}} \sim \mathcal{N}(0, \mathbf{I})$ is used across all training–query pairs to reduce stochastic variance. Gradients are projected from dimension $D = 1\,418\,996\,800$ to $D' = 512$ using a Fastfood John-

son–Lindenstrauss projection $\mathbf{P}$ selected via the search in Fig.4 to balance performance and storage. Motion weights $\mathbf{W}$ are computed from AllTracker flow magnitudes $M_f$, min–max normalized to $[0, 1]$ with a small bias $\zeta = 10^{-6}$. All computations use bfloat16 precision for memory efficiency.

**Subset Selection & fine-tuning.** For any number of query points, we select top-10% data of the datasets. We finetune the `Wan2.1-T2V-1.3B` backbone while freezing both the T5 text encoder (Raffel et al., 2020) and the VAE. The input resolution is fixed to $480 \times 832$ pixels. We use a learning rate of $1 \times 10^{-5}$ and the AdamW optimizer (Loshchilov & Hutter, 2017) following the DiffSynth-Studio defaults. We train the models for 1 epoch with the dataset repeated 50 times.

**Evaluation.** The test set consists of the same 10 motion categories, but with different visual appearances, compared with the query set. We provide the prompt samples below.

---

### Samples from Motion Query Set

We illustrate representative prompts from our query set used to generate query videos with Veo-3.

```
compress, "A slice of sandwich bread flattened by a flat metal plate,
steady camera, soft studio lighting, plain backdrop;
emphasize air pockets collapsing."

bounce, "A ping-pong ball bouncing on a white table,
steady side camera, neutral light, seamless backdrop;
emphasize consistent bounce height and timing."

roll, "A spool of thread rolling from left to right, close-up static
camera, bright studio light; highlight axle rotation and
smooth travel."

explode, "A single balloon bursting into fragments, captured in
high-speed slow motion with a fixed camera, bright even lighting,
seamless background; emphasize outward debris and air release."

float, "A foam cube floating on the surface of water,
static overhead camera, bright light, clean tank;
emphasize buoyancy and slight rocking."
```

---

### Samples from Motion Test Set

We illustrate representative prompts from our test set used to generate test videos with our finetuned models.

```
compress, "A rubber ball being compressed under a flat press,
filmed with a stationary camera. Bright, shadow-free lighting
and a clean background emphasize the deformation as it flattens."

bounce, "A basketball bouncing vertically on a wooden court plank,
unmoving camera, balanced indoor lighting, plain wall background;
clearly show deformation at impact."

roll, "A bike tire rolling freely on a stand, static side camera,
indoor neutral light; show uniform rotation without wobble."

explode, "A fragile glass ornament breaking apart mid-air,
fixed camera, bright controlled lighting, plain backdrop;
capture shards and reflections crisply."

float, "A green leaf floating gently on perfectly still water
in a transparent tank, fixed top-down camera, bright even lighting;
emphasize surface tension ripples."
```

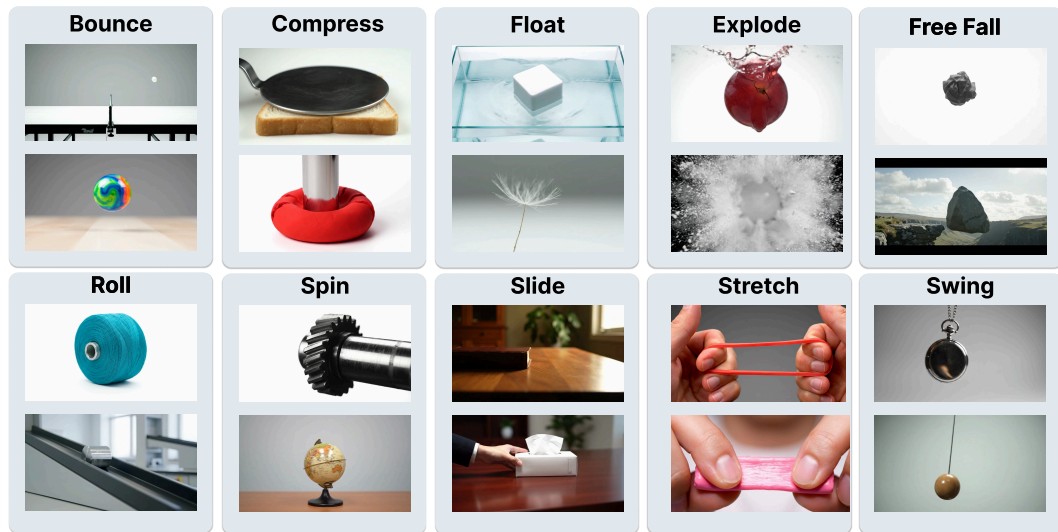

Figure 8: **Illustration of motion query set.** We generate near-realistic video queries with Veo-3 across ten motion categories. Each category contains five query videos synthesized with controlled prompts and manually screened for clarity and physical plausibility.

### F.2   DETAILS ON MOTION QUERY DATA

A small, controlled set of query videos is constructed to isolate specific motion primitives while minimizing confounding factors (e.g., textured backgrounds, uncontrolled camera motion). Such clean and consistent clips are challenging to obtain from natural data sources. To address this, we synthesize the query set using Veo-3 (Google DeepMind, 2025) and apply a strict post-generation screening for physical plausibility and generation realism. We target ten motion types: *compress, bounce, roll, explode, float, free fall, slide, spin, stretch, swing*. For each category, we retain 5 query samples, yielding a total of 50 queries. This scale provides adequate coverage of the motion taxonomy used in our evaluations while maintaining tractable attribution computation. We further provide a few examples of the generation prompts and the generated video query set in Fig. 8.

## G   USAGE OF LARGE LANGUAGE MODELS (LLMS)

During the preparation of this work, the authors used LLMs to support various aspects of the research and writing process. The specific applications of LLMs in this paper are outlined below:

**Code.** Helped generate/debug auxiliary Python for gradient, similarity, and visualization pipelines, and suggested fixes for training errors; authors validated all changes.

**Manuscript.** Assisted with LaTeX table/figure formatting and appendix templates.

**Writing.** Provided copy-editing (grammar, phrasing, brief summaries). All technical claims and interpretations were written and verified by the authors.

## H   DISCUSSION

### H.1   RUNTIME

We provide a detailed runtime breakdown from our experiments on 10k training samples with `Wan2.1-T2V-1.3B` model to address scalability concerns. The key insight is that the dominant cost of our pipeline is computing per-training-sample gradients, which is performed once and then can be reused for all subsequent queries. Each training clip's gradient is projected into a compact 512-dimensional vector, and adding a new query requires only (i) a single backward pass to obtain its own projected 512-dimensional gradient and (ii) computing cosine similarity between the query

| Component | Complexity | Runtime | Notes |
|---|---|---|---|
| Gradient computation | $\mathcal{O}(B)$ per sample | Query: $\sim$54 seconds
Training: $\sim$150 hours | 1 A100 GPU;
Single forward+backward pass; training is dominant cost but amortized over all queries;
embarrassingly parallel (with 64 GPUs, $\sim$2.3 hours) |
| Projection | $\mathcal{O}(\|\mathcal{D}\| \cdot D' \log D')$ | $\sim$1.97 seconds per sample | $D' = 512$ |
| Influence computation | $\mathcal{O}(\|\mathcal{D}\| \cdot D')$ | $\sim$46 milliseconds per query | 1 query $\times$ 10k training samples |
| Majority-vote aggregation | $\mathcal{O}(\|\mathcal{D}\| \cdot q)$ | $\sim$139 milliseconds | 50 queries $\times$ 10k samples |

Table 6: **Runtime Breakdown.** Detailed computational complexity and runtime for each component of our motion attribution framework on 10k training samples with `Wan2.1-T2V-1.3B` model.

vector and stored training vectors, which is exceptionally lightweight (on the order of seconds). Thus, the computational burden does not scale with the number of queries but only with the size of the training set. As shown in Tab. 6, the dominant cost is the one-time training data gradient computation ($\sim$150 hours on 1 A100), which is amortized across all queries. Once computed, adding a new query requires only $\sim$54 seconds (gradient computation) + 46ms (influence computation) = $\sim$54 seconds total. The training data gradient computation is embarrassingly parallel and can be reduced to $\sim$2.3 hours with 64 GPUs.

**Runtime comparison with baselines.** We compare the computational cost of our method with the baseline approaches for processing 10k training samples on a single GPU (Table 7):

| Method | Random | Motion Mag. | Optical Flow | V-JEPA | Ours |
|---|---|---|---|---|---|
| **Total for 10k (1 GPU)** | $<$1 second | $\sim$5.5 hours | $\sim$5.7 hours | $\sim$3 hours | $\sim$150 hours |

Table 7: **Runtime Comparison with Baselines.** Total computational time required for processing 10k training samples on a single GPU across different data selection methods.

While our method requires more upfront computation than the baseline approaches, this cost is amortized across all queries, and the computed gradients can be reused for multiple selection queries, making it practical for large-scale data curation scenarios.

## H.2 LIMITATIONS

Our analysis treats each video as a whole unit, which avoids collapsing motion into frame-level appearance, but risks overlooking the fact that only certain segments may carry motion-relevant information. Highly informative intervals can be diluted when averaged with static or redundant portions of the same clip. This suggests an open direction toward finer-grained attribution at the motion segment or motion event level, which could reveal more precise insights into how different phases of a trajectory shape motion learning. Another limitation is that our motion masks may overemphasize camera-only motion; we detect this by spatial uniformity of $\mathbf{W}$ and down-weight such clips, but a full disentanglement of ego and object motion remains future work.

Additionally, our framework does not explicitly account for classifier-free guidance (CFG), which is widely used in practice to steer video generation but introduces discrepancies between training-time attribution and inference-time dynamics. As a result, our influence estimates may not fully capture how guidance alters motion behavior. In addition, while attribution-driven fine-tuning improves targeted motion quality, it may introduce trade-offs with base model capabilities. This raises the need for future work on balancing targeted motion adaptation with the preservation of broader generative capabilities.

### H.3 FUTURE DIRECTIONS

**Tracker-robust motion saliency.** Replace or ensemble AllTracker with alternative estimators and use its confidence/visibility channels to weight masks.

**Closed-loop data curation.** Move from one-shot ranking to active selection: iteratively attribute, finetune, and re-attribute or replace simple majority voting with learned query weights.

**Safety and governance.** Use negative-influence filtering to suppress undesirable or unsafe dynamics, document curator choices, and audit motion behaviors exposed by our framework.

**Self-generated video queries.** Use model-generated videos as queries to trace problematic motion patterns (e.g., unrealistic physics) back to training data, enabling iterative diagnostics and targeted motion improvement.

## I VISUALIZATION

In this section, we provide additional visualizations to demonstrate the effectiveness of our method.

### I.1 MOTION VISUALIZATION

To provide intuition for the behavior of our motion-weighted loss, we visualize the motion magnitude patch. We compute per-pixel motion magnitude using optical flow and apply motion-based weighting that preserves the appearance of dynamic regions while attenuating static backgrounds. This centered gray visualization directly illustrates the spatial weighting applied by our motion loss during training.

Fig. 9 presents representative frames from our dataset, comparing original frames with their corresponding centered gray visualizations for eight distinct video samples. These visualizations show that the motion-weighted loss preferentially emphasizes dynamic content while down-weighting static scene elements.

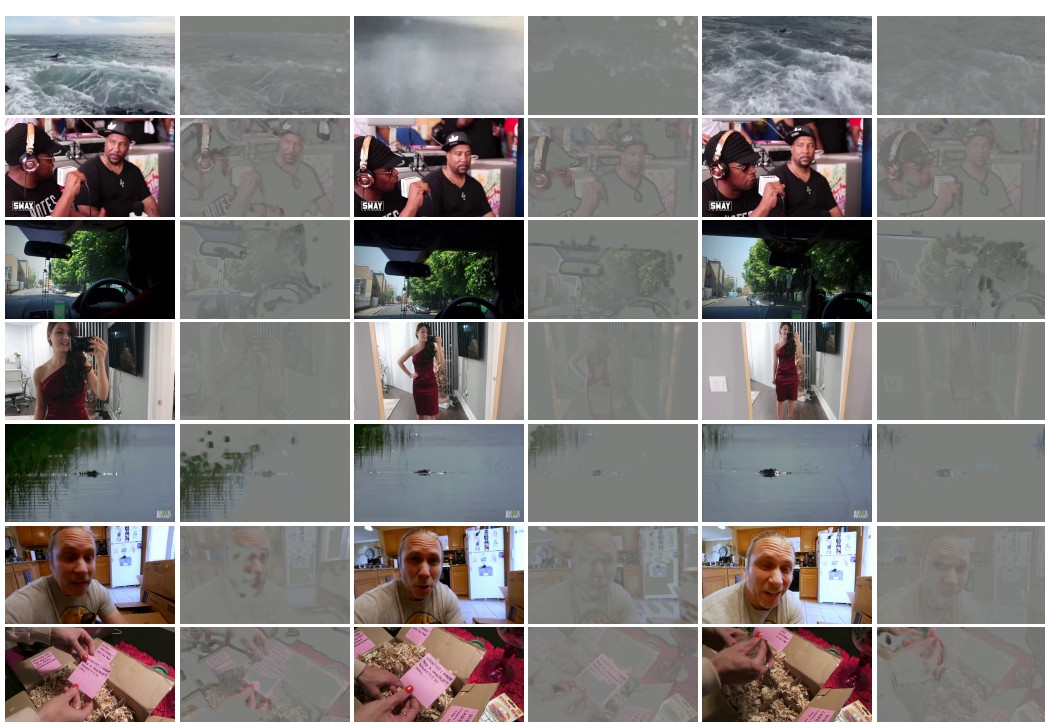

Figure 9: **Motion-weighted visualization.** Comparison of original frames and centered gray visualizations for eight video samples across three time points (early, middle, late). The centered gray visualization demonstrates the spatial weighting applied by our motion loss: dynamic regions remain visible while static backgrounds are attenuated to neutral gray. *Takeaway:* The provides heuristic intuition into what information our motion attribution focuses on – the information in grayer regions, which lack motion, is down-weighted by our method.

