# OpenReview forum: "Where is Motion From? Scalable Motion Attribution for Video Generation Models"
_ICLR.cc/2026/Conference — Submitted to ICLR 2026_

### Official Review · Reviewer_PTzY · 2025-10-28

**Soundness:** 3
**Presentation:** 3
**Contribution:** 2
**Rating:** 4
**Confidence:** 4

**Summary:**

This paper proposes selecting a subset of motion-influential data for training video generative models. The goal is to choose data that strongly affects motion and to guide data curation in a way that improves temporal consistency and physical plausibility. A computational method is used to compare the gradients of query and training data points for this purpose.

**Strengths:**

1. The paper is well-written and easy to follow.

2. Both computational experiments and human studies are provided to support the claims.

**Weaknesses:**

**Scalability concern:**  In the experiment, there are 50 query videos and 10k clips in the training subset. Adding just one more query requires computing influence scores 10k additional times. How to address this limit?

**Equation (5) issue:** How can the exact cardinality of the sampled timestep-and-noise set \\( \\mathcal{T} \\) be computed? There are infinitely many choices for the noise vector and time step in the flow matching framework. Shouldn’t it be better represented as an expectation?

**Query data instance:** Can we use a self-generated video and what does it imply?

**Equation (7) parameter:**   fixing $\\epsilon_{\\text{fix}}$ is sort of understandable. But why is the timestep $t_{\\text{fix}}$ fixed?

**Equation (8) – Structured projection:** Why select the structured projection in this way? Many terms involve random operators; why is a statistical operator (e.g., expectation) not needed?

**Comparison metrics:**  Suggest using other metrics such as FVMD for more comprehensive quantitative results [1].

**Human evaluation experiment:**  Only 20 videos are used (10 from a baseline, 10 from the proposed method).  This seems too few to yield convincing results.


Ref:\
[1] Liu, J., Qu, Y., Yan, Q., Zeng, X., Wang, L. and Liao, R., 2024. Fr\'echet Video Motion Distance: A Metric for Evaluating Motion Consistency in Videos. arXiv preprint arXiv:2407.16124.

**Questions:**

Please see the [Weakness] section.

---

> ### Author Response · Authors · 2025-11-24
>
> We thank the reviewer for the detailed feedback and address each concern below.
>
> **Scalability concern.** We appreciate this concern and we want to clarify that the dominant cost of our pipeline is computing per-training-sample gradients, which is performed once and then can be reused for all subsequent queries. Each training clip's gradient is projected into a compact 512-dimensional vector, and adding a new query requires only (i) a single backward pass to obtain its own projected 512-dimensional gradient and (ii) compute cosine similarity between the query vector and stored training vectors, which is extremely lightweight (on the order of seconds). Thus, the computational burden does not scale with the number of queries but only with the size of the training set. For clarity, we summarize the complexity of each component below, following the notations in Sec. 3.6:
>
> | Component | One-time or Per-query? | Complexity | Notes |
> |-----------|------------------------|------------|-------|
> | Training-data gradients | One-time | O(\|D\| · B) | Dominant cost; reused for all queries |
> | Projection | One-time | O(\|D\| · D' log D') | D' = 512; negligible vs. backward pass |
> | Query gradient | Per-query | O(B) | Single backward pass |
> | Influence computation | Per-query | O(\|D\| · D') | 10k × 512 dot products |
> | Majority-vote aggregation | Per-set | O(\|D\| · Q) | Thresholding precomputed scores |
>
>
> **Equation (5) issue.** The |T| terms in Eq. (5) refer to the number of samples used to approximate the expectation over timesteps and noise rather than the cardinality of the underlying continuous domain, following standard practice in the diffusion/flow-matching literature. In diffusion/flow-matching training, the continuous expectation over (t, ε) is always approximated using a finite number of sampled timestep-noise pairs. Our notation in Eq. (5) matches the formulation used in the previous literature [1], where the sample-average estimator is used.
>
> **Query data instance.** Yes, that is a great point! Self-generated videos can be used as queries. This is particularly useful for iterative improvement: one can generate videos with the current model, identify specific motion patterns that need improvement (e.g., unrealistic physics), and our approach can serve as a diagnostic tool to trace the self-generated videos back to the training data that caused problematic generations and enabling targeted motion improvement based on observed failure modes. We will add a discussion of this use case in the revised manuscript.
>
> **Equation (7) parameter.**  In practice, we found that using t=500 (middle of the denoising process) provides a good approximation (See Sec. 4.4 ablation on Single-timestep attribution). However, one could also aggregate influence scores across multiple timesteps if desired, at a cost of additional computation.
>
> **Equation (8) – Structured projection.** The Fastfood transform is a deterministic structured random projection that uses random operators constructed once and then fixed. The randomness is in the construction of the projection matrix, not in its application. The projection matrix P is fixed after initialization and this is why no expectation operator is needed. The projection provides computational efficiency while preserves inner products with high probability [2].
>
>
> **Comparison metrics.** We thank the reviewer for suggesting FVMD metric and we have added FVMD evaluation in the table below:
>
> FVMD Evaluation Results:
> | Method | Base | Full FT | Random | Motion Mag. | Optical Flow | V-JEPA | Ours w/o M. Masking | Ours |
> |--------|------|---------|--------|-------------|--------------|--------|-------------|------|
> | FVMD (↓) ×10³ | 7.325 | 5.837 | 7.297 | 6.033 | 6.654 | 5.975 | 5.846 | **5.628** |
>
> Lower FVMD indicates better motion consistency. All baseline approaches and our MOTIVE select 10% of the training data. Note that we added three additional baselines as per Reviewer 8yPp's suggestion. The results show that our approach improves motion consistency.
>
> **Human evaluation experiment.** We have expanded our human evaluation experiments. We now evaluate 50 videos with 17 participants, resulting in 850 total pairwise comparisons. The expanded evaluation aligns with our findings, showing strong human preference for our method compared to the baselines:
>
> Expanded Human Evaluation Results:
> | Method | Win (%) | Tie (%) | Loss (%) |
> |--------|---------|---------|----------|
> | Ours vs. Base | 74.1 | 12.3 | 13.6 |
> | Ours vs. Random | 58.9 | 12.1 | 29.0 |
> | Ours vs. Full FT | 53.1 | 14.8 | 32.1 |
> | Ours vs. w/o Motion masking | 46.9 | 20.0 | 33.1 |
>
>
> [1] Jinxu Lin, Linwei Tao, Minjing Dong, and Chang Xu. Diffusion attribution score: Evaluating training
> data influence in diffusion models. arXiv preprint arXiv:2410.18639, 2024.
>
> [2] Johnson, William B., and Joram Lindenstrauss. "Extensions of Lipschitz mappings into a Hilbert space." Contemporary mathematics 26.189-206 (1984): 1.

---

> > ### Comment · Reviewer_PTzY · 2025-11-24
> >
> > Thanks for your efforts in providing additional results in the rebuttal. I am still curious about the actual runtime and scalability. Could you provide a runtime breakdown of each step from real experiments (measured in milliseconds or seconds) and, if convenient, include a comparison with the baselines? This would offer more insight for practitioners to evaluate how they might adopt this method and potentially increase its impact.

---

> > > ### Author Response · Authors · 2025-11-25
> > >
> > > We thank the reviewer for the prompt and helpful reply! We provide a detailed runtime breakdown from our experiments on 10k training samples with Wan2.1-T2V-1.3B model.
> > >
> > > | Component | Complexity | Runtime | Notes |
> > > |-----------|------------|---------|-------|
> > > | Query data gradient computation | O(B) | ~54 seconds per query | Single forward+backward pass |
> > > | Training data gradient computation | O(\|D\| · B) | ~150 hours (for 10k samples on 1 A100 GPU, 54s/sample) | Dominant cost but amortized over all queries and runs; embarrassingly parallel (with 64 GPUs, ~2.3 hours) |
> > > | Projection | O(\|D\| · D' log D') | ~1.97 seconds per sample | D' = 512; negligible vs. backward pass |
> > > | Influence computation | O(\|D\| · D') | ~46 milliseconds per query | 1 query × 10k training samples, ~4.6 μs per query-training pair |
> > > | Majority-vote aggregation | O(\|D\| · Q) | ~139 milliseconds | 50 queries × 10k samples (44ms voting + 94ms aggregation + 1ms sorting) |
> > >
> > > As shown in the table above, the dominant cost is the one-time training data gradient computation (~150 hours on 1 A100), which is amortized across all queries. Once computed, adding a new query requires only ~54 seconds (gradient computation) + 46ms (influence computation) = ~54 seconds total.
> > >
> > > **Runtime comparison with baselines:**
> > >
> > > | Method | Random | Motion Mag. | Optical Flow | V-JEPA | Ours |
> > > |--------|--------|-------------|--------------|--------|------|
> > > | Total for 10k (1 GPU) | <1 second | ~5.5 hours | ~5.7 hours | ~3 hours | ~150 hours |
> > >
> > > We will include this runtime analysis in the revised manuscript and we are eager to answer any additional concerns or questions you may have.

---

> > > > ### Comment · Reviewer_PTzY · 2025-11-25
> > > >
> > > > Thank you for your response and for being transparent about the performance differences. While I truly appreciate your efforts and dedication, I still have concerns about the scalability of large-scale applications given the numbers shared. Based on our overall discussion, I’d like to raise my score to 6 and will wait for the other reviewers'/AC's updates and feedback.

---

> > > > > ### Author Response · Authors · 2025-11-25
> > > > >
> > > > > Thank you so much for the thoughtful follow-up and for raising the score, we really appreciate it! We’re motivated by the positive feedback and look forward to the other reviewers’ and AC’s comments.

---

### Official Review · Reviewer_MAhn · 2025-10-30

**Soundness:** 3
**Presentation:** 3
**Contribution:** 3
**Rating:** 8
**Confidence:** 3

**Summary:**

This paper addresses the underexplored issue of motion attribution in video generative models by proposing MOTIVE, a scalable gradient-based framework. Existing methods fail to separate motion from static appearance and lack scalability for large datasets/models. MOTIVE tackles this via three key steps: using flow-weighted masks to isolate temporal dynamics, correcting frame-length bias for fair scoring, and applying Fastfood projection for efficient gradient storage/computation. It calculates motion influence scores for training clips, selects top 10% high-impact data for finetuning. Experiments on VIDGEN-1M/4DNeX-10M with Wan2.1-T2V-1.3B show MOTIVE outperforms baselines: 89.4% dynamic degree on VBench (surpassing full-dataset finetuning’s 84.7%) and 76.7% human preference win rate vs. pretrained models, proving its value for targeted data curation and motion quality improvement.

**Strengths:**

1. Critical and Well-Targeted Problem FormulationThe paper focuses on a pivotal, underexplored gap in video generative models: identifying training data that drives motion quality (a core video feature distinct from static appearance). Filtering high-quality motion data is vital for finetuning—where carefully selected clips significantly boost temporal coherence and physical plausibility—filling a key need for practical video model optimization.

2. Intuitive and Principled Method DesignMOTIVE’s approach is highly logical and video-specific. It isolates motion from static content via flow-weighted loss masks (using optical flow for dynamic regions), corrects frame-length bias (avoiding spurious long-clip ranking), and uses Fastfood projection for scalability. These choices directly fix image-centric attribution limits, making the framework theoretically sound and practically feasible.

3. Comprehensive and Rigorous ExperimentsExperiments are thorough: evaluations on VIDGEN-1M/4DNeX-10M, VBench for motion metrics, diverse baselines (random selection, full finetuning), and human evaluations (76.7% preference vs. base model). Ablations (single-timestep validity, projection dimension impact) validate components, ensuring robust, credible conclusions.

**Weaknesses:**

All experiments in the paper are exclusively conducted on the Wan2.1-T2V-1.3B model, with no validation on other mainstream video generative architectures (e.g., 3D U-Nets, latent video VAEs with different temporal attention blocks, or non-DiT-based diffusion models). As the paper itself acknowledges, "our evaluation centers on one open-source backbone due to compute; broader portability is future work" — this single-model focus means the framework’s effectiveness, such as motion mask compatibility, gradient projection stability, and finetuning gain consistency, cannot be confirmed for other popular video diffusion models. This limitation reduces confidence in the framework’s general applicability to diverse video generation systems, weakening the persuasiveness of its universal utility.

**Questions:**

This paper addresses a crucial problem in data of video diffusion training with detailed analysis, making it worthy of acceptance. As it only tests Wan2.1-T2V-1.3B, adding one more model test would further boost its quality.

---

> ### Author Response · Authors · 2025-11-24
>
> We sincerely thank the reviewer for the encouraging feedback and valuable suggestions!
>
> Our method is architecture-agnostic. Motion masking operates at the loss-function level and does not require model-specific modifications. The gradient projection via Fastfood transform applies to any model parameter space, and the influence computation works with any gradient-based training procedure. We have added experiments on Wan2.2-TI2V-5B which introduces a much larger parameter count (5B vs. 1.3B) and a new high-compression Wan2.2-VAE, to test the generalizability of our framework, and the results below show that our approach generalizes effectively to different model designs.
>
> VBench Evaluation on Wan2.2-TI2V-5B
> | Method | Subject Consist. | Background Consist. | Motion Smooth. | Dynamic Degree | Aesthetic Quality | Imaging Quality |
> |--------|-----------------|---------------------|----------------|----------------|-------------------|-----------------|
> | Base | 94.9 | 96.4 | 97.5 | 84.0 | 44.4 | 65.5 |
> | Full fine-tuning | **95.3** | 96.5 | 97.5 | 87.5 | 44.8 | 66.2 |
> | Random selection | 94.7 | 96.2 | 97.3 | 83.8 | 44.6 | 65.2 |
> | Ours w/o motion masking | 94.9 | 96.5 | 97.4 | 86.2 | 45.2 | 64.8 |
> | **Ours (MOTIVE)** | 95.1 | **96.6** | **97.6** | **91.0** | **45.6** | 65.5 |

---

> > ### Comment · Reviewer_MAhn · 2025-11-28
> > **Comment**
> >
> > Thanks for the author's new experiments, it address my concern, I will keep my original score.

---

### Official Review · Reviewer_FHs6 · 2025-10-31

**Soundness:** 2
**Presentation:** 2
**Contribution:** 2
**Rating:** 2
**Confidence:** 4

**Summary:**

This paper investigates the under-explored problem of attributing motion in generated videos to specific training clips, introducing MOTIVE, a gradient-based, motion-aware data attribution framework for video diffusion models. The key idea is to isolate temporal dynamics from static appearance by re-weighting gradients with flow-derived motion masks, enabling scalable influence estimation over modern billion-parameter models. Extensive experiments on VIDGEN-1M and 4DNeX-10M show great performance of MOTIVE.

**Strengths:**

- This paper opens an important new topic—motion-centric data attribution—that prior image-oriented methods cannot address.

- The authors propose a simple yet effective idea: flow-weighted gradient masking that disentangles motion from appearance without changing the forward generative process.

- Experiments are good, including large-scale datasets and human evaluations.

**Weaknesses:**

I would like to begin by setting aside the specific technical details and share my perspective on the problem that this paper aims to address. Based on my research experience in video generation, I have observed a clear trade-off between motion dynamics and the occurrence of visual artifacts—in general, stronger motion dynamics tend to correlate with a higher probability of artifacts. I believe this trade-off represents one of the most fundamental challenges in current video generation research. From the experimental results presented in the paper (e.g., Table 1), it appears that the proposed method still suffers from this dilemma. In other words, while the approach seems to improve the Dynamic Degree metric significantly, this improvement may come at the cost of other important aspects such as Background Consistency and Imaging Quality. If the main contribution of this work is limited to enhancing motion dynamics at the expense of overall visual stability and quality, such improvements could arguably be achieved more simply by using training data with inherently higher motion. Overall, I appreciate the authors’ motivation to address the problem of video dynamism. However, the current method seems not to fundamentally resolve the underlying trade-off and not to tackle the core technical challenges of dynamic yet consistent video generation.

In addition, I believe the paper has the following shortcomings:
(1) Writing quality—the overall writing could be improved. For example, each equation should be properly punctuated, and the presentation could benefit from more polished academic writing.
(2) Experimental sufficiency—the experiments are relatively limited, as they are conducted only on the Wan-2.1 model. This raises concerns about the reliability and generalizability of the results. It would be more convincing if the authors could include additional models, such as HunyuanVideo, to validate the effectiveness and robustness of their approach.

**Questions:**

see weakness

---

> ### Author Response · Authors · 2025-11-24
>
> We thank the reviewer for their thoughtful feedback and address each concern below.
>
> **Our method does not simply trade visual quality for motion dynamics.** We acknowledge the concern that our method "may come at the cost of other important aspects such as Background Consistency and Imaging Quality." However, our VBench results (Table 1) show that while Background Consistency decreases modestly (-0.5% vs. full finetuning), four out of six metrics actually improve or maintain the same performance: Subject Consistency (+0.4%), Motion Smoothness (identical), Aesthetic Quality (+1.0%), and Dynamic Degree (+4.7% to 89.4%). Notably, our Imaging Quality (64.6%) is actually higher than full finetuning (63.9%). This shows motion improvements while maintaining or improving most quality metrics, rather than simply trading quality for motion.
>
> Furthermore, it is important to note that evaluating video generation quality remains a long-standing challenge in the field, as there is no standard or perfect benchmark. Human evaluation is widely recognized as the gold standard for evaluating the generation quality. Our human evaluation results show that our approach achieves a 57.5\% preference win rate compared to the full finetuning. This shows a limitation of current evaluation benchmarks: the visual quality decreases captured by automatic metrics such as background consistency are not the dominant factor in how humans judge generated video quality.
>
> **Experiments with additional architectures.** We acknowledge this concern and we further test our framework on additional video generation architectures beyond Wan2.1-T2V-1.3B. We have included the experiment results of MOTIVE on Wan2.2-TI2V-5B which introduces a much larger parameter count (5B vs. 1.3B) and a new high-compression Wan2.2-VAE, and it shows that our approach generalizes effectively to different model designs.
>
> VBench Evaluation on Wan2.2-TI2V-5B
> | Method | Subject Consist. | Background Consist. | Motion Smooth. | Dynamic Degree | Aesthetic Quality | Imaging Quality |
> |--------|-----------------|---------------------|----------------|----------------|-------------------|-----------------|
> | Base | 94.9 | 96.4 | 97.5 | 84.0 | 44.4 | 65.5 |
> | Full fine-tuning | **95.3** | 96.5 | 97.5 | 87.5 | 44.8 | 66.2 |
> | Random selection | 94.7 | 96.2 | 97.3 | 83.8 | 44.6 | 65.2 |
> | Ours w/o motion masking | 94.9 | 96.5 | 97.4 | 86.2 | 45.2 | 64.8 |
> | **Ours (MOTIVE)** | 95.1 | **96.6** | **97.6** | **91.0** | **45.6** | 65.5 |
>
>
>
> **Writing quality improvements.** We are currently in the process of revising the manuscript and will update it as soon as possible. We would greatly appreciate more specific pointers on which sections could be further improved and we will revise the paper accordingly.

---

### Official Review · Reviewer_8yPp · 2025-11-01

**Soundness:** 3
**Presentation:** 3
**Contribution:** 2
**Rating:** 4
**Confidence:** 2

**Summary:**

This paper introduces MOTIVE, a framework for motion-centric data attribution in video diffusion models. While existing attribution methods primarily analyze static appearance in image diffusion, MOTIVE aims to identify which finetuning clips most influence temporal dynamics in generated videos. The key idea is to compute motion-weighted gradients, where optical-flow–based masks emphasize dynamic regions while suppressing static backgrounds.

**Strengths:**

The paper identifies an underexplored but meaningful question: “Which training data drives motion learning in video diffusion models?” This focus on motion attribution provides a clear conceptual step beyond standard image-level attribution, and the method effectively extends existing data attribution frameworks to the video domain with appropriate modifications for temporal structure and scalability.

**Weaknesses:**

**Limited evaluation and unclear attribution advantage**
The proposed attribution-based selection may in practice act as a proxy for identifying motion-rich or high-quality clips, rather than truly capturing data that causally influences motion learning. This concern is amplified by the evaluation setup, where the method is compared only against random selection, a trivial baseline that cannot disentangle whether improvements stem from genuine attribution or simply from favoring dynamic, well-captured videos. A more meaningful comparison would involve finetuning with datasets selected by explicit motion-quality criteria, such as average motion magnitude, optical-flow statistics, or reward-model scores reflecting motion realism or physical plausibility. Without such baselines, it remains unclear whether the proposed approach provides any advantage beyond straightforward motion-saliency filtering.

**Questions:**

See the weakness.

---

> ### Author Response · Authors · 2025-11-24
>
> We thank the reviewer for this valuable feedback. We have now added additional analysis and extensive baseline comparisons to address this concern.
>
> **MOTIVE is not simply selecting "motion-rich" clips:** The key distinction is that our influence scores are computed via gradients, and training videos are considered influential only when they directly improve the model's ability to generate the target motion dynamics, not because they contain more motion overall.
>
> To empirically validate this, we further analyze the distribution of motion magnitudes in our selected data. We compute the mean motion magnitude for 10k videos in the VIDGEN dataset and compare the distributions of our top 10% selected videos (highest influence scores) with those of the bottom 10% (lowest influence scores).
>
> Our top 10% selected videos have a mean motion magnitude of 3.85, which is only 4.3% higher than the bottom 10% (3.69), despite representing opposite extremes of influence scores. The analysis also shows that in the moderate motion range (bins 3, 4, and 5), the top 10% of positive-influence samples outnumber the bottom 10% of negative-influence samples, yet both groups also appear in low-motion bins (0-2) and high-motion bins (6-9).
>
> This distribution pattern shows that high-influence videos selected by MOTIVE span the entire motion spectrum, not just high-motion regions. Many high-motion videos receive low influence scores, while numerous influential videos exhibit modest or even low motion magnitude. These findings show that our motion attribution approach captures training influence, focusing on motion rather than simply acting as a motion-saliency filter. We include this analysis and figure in Appendix Sec. D.1 of the revised manuscript.
>
> **Additional baseline experiments:** Following the reviewer's suggestion, we have added three additional baselines that explicitly select data based on motion-quality criteria:
>
> (1) *Motion magnitude*: selects videos with the highest average motion magnitude;
>
> (2) *Optical flow statistics*: selects videos with the highest composite scores based on motion magnitude, motion entropy, spatial coverage, motion complexity, and temporal consistency;
>
> (3) *V-JEPA embeddings*: selects videos that are most representative of motion patterns using self-supervised spatiotemporal V-JEPA 2 features, which capture high-level motion semantics.
>
>
>
> ### VBench Evaluation Results
>
> | Method | Subject Consist. | Background Consist. | Motion Smooth. | Dynamic Degree | Aesthetic Quality | Imaging Quality |
> |--------|-----------------|---------------------|----------------|----------------|-------------------|-----------------|
> | Base | 95.3 | 96.4 | 96.3 | 82.3 | 45.3 | **65.7** |
> | Full fine-tuning | 95.9 | **96.6** | 96.3 | 84.7 | 45.0 | 63.9 |
> | Random selection | 95.3 | 96.6 | 96.3 | 81.6 | 45.7 | 65.1 |
> | Motion magnitude | 95.6 | 96.2 | 95.7 | 82.0 | 45.1 | 63.2 |
> | Optical flow | 95.3 | 96.4 | 95.3 | 80.0 | 45.4 | 62.9 |
> | V-JEPA embed. | 95.7 | 96.0 | 95.6 | 82.0 | 44.9 | 62.7 |
> | Ours w/o MM | 95.4 | 96.1 | 96.3 | 85.3 | 45.7 | 63.2 |
> | **Ours (MOTIVE)** | **96.3** | 96.1 | 96.3 | **89.4** | **46.0** | 64.6 |
>
> Vbench results in the table above show that our attribution-based method consistently outperforms these motion-quality criteria based approaches. We include the additional baseline results (motion magnitude, optical flow, and V-JEPA embeddings)  in the Tab.1 of the revised paper.

---

### Author Response · Authors · 2025-11-28
**Thank you!**

We deeply appreciate the Area Chairs taking on this paper under challenging circumstances following the recent ICLR incident. We understand the additional workload this creates for Area Chairs, and to help navigate the substantial volume of feedback and our detailed responses, we provide this summary highlighting the key strengths recognized by reviewers, the main concerns raised, and how we have thoroughly addressed each one.

We sincerely thank all reviewers for their thoughtful and constructive feedback. We are delighted that reviewers found our paper addresses "a pivotal, underexplored gap in video generative models" (Reviewer MAhn), "an underexplored but meaningful question" (Reviewer 8yPp), and "opens an important new topic—motion-centric data attribution—that prior image-oriented methods cannot address" (Reviewer FHs6). We are motivated by comments that our work features "intuitive and principled method design" and "a simple yet effective idea" with "comprehensive and rigorous experiments" (Reviewers MAhn, FHs6), that our paper is "well-written and easy to follow" (Reviewer PTzY), and that our method "effectively extends existing data attribution frameworks to the video domain with appropriate modifications for temporal structure and scalability" (Reviewer 8yPp).

All reviewers offered constructive criticism, which we have used to substantially improve our paper. During the rebuttal period, we provided detailed responses to each concern, which involved conducting extensive experiments. **We are grateful that two reviewers engaged with the discussion: Reviewer PTzY raised their score to 6 after our responses, and Reviewer MAhn maintained their score of 8.** While we did not hear back from the other reviewers given the limited time after the ICLR incident, we believe our revisions thoroughly address their concerns as well.

**Summary of Revisions**

We incorporate the following major additions to the revision (highlighted in blue in the revised manuscript to help ACs & reviewers focus on key changes; note that we have also made various minor writing improvements, figure adjustments, and clarifications throughout the paper that are not highlighted):

1. Additional model experiments (Reviewers FHs6, MAhn): Experiments on Wan2.2-TI2V-5B, confirming generalizability across different video generation architectures.
2. Additional baseline comparisons (Reviewer 8yPp): Three motion-quality baselines (motion magnitude, optical flow statistics, V-JEPA embeddings), demonstrating our attribution-based approach outperforms motion-saliency filtering.
3. Additional evaluation metrics (Reviewer PTzY): FVMD (Fréchet Video Motion Distance) evaluation for comprehensive quantitative assessment of motion consistency.
4. Expanded human evaluation (Reviewer PTzY): Increased scale with more videos and participants for statistically robust preference assessments.
5. Motion distribution analysis (Reviewer 8yPp): Analysis showing selected videos span the entire motion spectrum (4.3% difference in mean motion magnitude between top and bottom 10%), confirming we capture motion influence rather than motion richness.
6. Technical clarifications & writing improvements (Reviewers PTzY, FHs6): Detailed explanations of computational scalability and self-generated video queries; revised equation punctuation and polished presentation throughout.

We sincerely thank the Area Chairs and all reviewers for their time, effort, and thoughtful engagement with our work, especially under these challenging circumstances. We are grateful for the opportunity to address the concerns and we hope the revisions show our commitment to producing impactful research for the community.

---

### Meta-Review · Area_Chair_mwzt · 2026-01-07

**Summary:**

This paper proposes MOTIVE, a motion-aware, gradient-based data attribution framework for video diffusion models that isolates temporal dynamics using flow-based masks and identifies training clips most influential to motion. The work received highly mixed scores (8,4,4,2). Most reviewers acknowledge that the problem studied is novel and meaningful, while major concerns include that the current method does not fundamentally address the underlying trade-off or core technical challenges of generating dynamic yet consistent videos, insufficient comparative experiments, lack of validation across diverse video models, and questions regarding scalability for large-scale applications.

**Reviewer Concerns:**

Concerns Likely Addressed in the Authors’ Response:

- Additional baseline comparisons: Three motion-quality baselines (motion magnitude, optical flow statistics, V-JEPA embeddings).
- Additional evaluation metrics and expanded human evaluation.
- Writing.

Concerns Partially Addressed:

- Almost all reviewers pointed out that conducting experiments only on Wan2.1-T2V-1.3B is insufficient to demonstrate the generality of the method. In the rebuttal, the authors provided results on Wan2.2-TI2V-5B. More comprehensive results across different backbones are still needed.

- The method may come at the cost of other important aspects, such as background consistency and imaging quality.


Concerns Still Likely Outstanding:

- Concerns about the scalability of large-scale applications.

**Reviewer Scores:**

Reviewer 8yPp

Original score: 4: marginally below the acceptance threshold. But would not mind if paper is accepted

Likely change: 6 (Marginally above acceptance threshold) or 4 (unchanged)

Reason: The reviewer’s main concern was the insufficient comparison with baselines; in the rebuttal, the authors provided three additional types of baselines, which partially alleviate this concern.

Reviewer FHs6

Original score: 2 (reject, not good enough)

Likely change: 2 (unchanged)

Reason: This reviewer’s main concern is the trade-off between motion dynamics and the occurrence of visual artifacts, noting that the proposed method still suffers from this dilemma. Another significant concern is the method’s generalization. In the rebuttal, the authors partially acknowledge minor losses in background consistency and provide additional results on Wan2.2-5B. However, since the underlying concerns are not fundamentally addressed and the tested backbone is limited and similar to Wan2.1, there is little willingness to adjust the score.

Reviewer MAhn

Original score: 8 (Accept)

Likely change: 8 (unchanged)

Reason: This reviewer already rated the paper highly. The authors’ rebuttal provided results on additional backbones, although the coverage is still limited.

Reviewer PTzY

Original score: 4: marginally below the acceptance threshold. But would not mind if paper is accepted

Likely change: 6 (Marginally above acceptance threshold)

Reason: After multiple rounds of discussion, the authors have addressed most of this reviewer’s concerns, although the reviewer remains concerned about the scalability of large-scale applications.

---

### Decision · Program_Chairs · 2026-01-26

Reject